



# CLIMCAPS Observing Capability for Temperature, Moisture and Trace Gases from AIRS/AMSU and CrIS/ATMS

Nadia Smith[1] and Christopher D. Barnet[1]

[1]Science and Technology Corporation, Columbia, Maryland, 212046, USA

*Correspondence to*: Nadia Smith (nadias@stcnet.com)

**Abstract.** The Community Long-term Infrared Microwave Combined Atmospheric Product System (CLIMCAPS) retrieves vertical profiles of temperature, water vapor, greenhouse- and pollutant gases as well as cloud properties from measurements made by infrared and microwave instruments on polar-orbiting satellites. These are AIRS/AMSU on Aqua and CrIS/ATMS on Suomi-NPP as well as NOAA20. CLIMCAPS retrieves these atmospheric soundings from each satellite platform

individually that together form a multi-platform, multi-instrument record of satellite soundings spanning nearly two decades of daily observations (2002 to present). Such a record is useful for characterizing diurnal and seasonal atmospheric processes from different time periods and regions across the globe. The strength of the observed signal at each scene is affected by a range of uncertainty sources broadly stemming from two classes, namely (i) the observing system (e.g., instrument type and noise, choice of inversion method, algorithmic implementation and assumptions) and (ii) localized conditions (e.g., presence

of clouds, rate of temperature change with pressure, amount of water vapor and surface type). If uncertainty is not explicitly quantified and reported, then retrieval products lose value in science and applications because their measured signals are obscured and, worse, misinterpreted. In this paper we characterize the CLIMCAPS Version 2.0 system and diagnose its observing capability for seven retrieval variables – temperature, $H_2O$, $CO$, $O_3$, $CO_2$, $HNO_3$ and $CH_4$ – from two satellite platforms, Aqua and NOAA20. We illustrate how CLIMCAPS observing capability varies spatially, from scene to scene and

latitudinally across the globe. We conclude with a discussion of how CLIMCAPS uncertainty metrics can be used in diagnosing its retrievals to promote understanding of the observing system as well as the atmosphere it measures.

## 1 Introduction

Instruments onboard satellites observe the global Earth atmosphere with unprecedented regularity in space and time. For any given scene on Earth today there are multiple observations from a range of different instruments measuring any number of

atmospheric variables. Our ability to monitor and characterize Earth systems is no longer limited by measurement scarcity but rather by our ability to process, interpret, apply and compare measurements from disparate sources with accuracy and consistency (Smith et al., 2013). Two space-based systems may observe the same atmospheric variable but at different view angles, different times of day, at different spatial or spectral resolutions measuring different aspects of the Earth's atmosphere. The challenge in inter-comparing different sources of remote observations is well documented (Stubenrauch et al., 1999;





Rodgers and Connor, 2003; Wylie et al., 2005; von Clarmann and Grabowski, 2007; Smith et al., 2013, 2015; Hearty et al., 2014; Gaudel et al., 2018). Straightforward side-by-side comparisons of disparate data sets can fail to yield meaningful insights because their differences cannot be explained by natural variability or instrument capability alone. Uncertainty masks the measured signal. Only with rigorous quantification and deliberate propagation of uncertainty through all data processing steps can a degree of transparency in observations be achieved so that the measured signal can be distinguished, uncertainty can be

characterized and data set differences be understood (Pougatchev et al., 1996; Ceccherini et al., 2003; Pougatchev, 2008; Ceccherini and Ridolfi, 2010; Hulley et al., 2012; Xiong et al., 2013; Merchant et al., 2017, 2019).

Pougatchev (2008) classified uncertainty in remote observations into two primary sources; namely (i) "state noncoincidence", or scene-dependent, effects such as spatial heterogeneity and temporal variation as well as, (ii) "characteristic differences", or observing system effects such as spectral resolution, footprint size and retrieval algorithm design. Uncertainty,

irrespective of its source, can be random (unreproducible) or systematic (reproducible). Random uncertainty can average out when data is aggregated, but systematic uncertainty propagates through analysis steps and obscures the measured signal in final results (Smith et al., 2015). It is, therefore, imperative to characterize systematic uncertainty as rigorously as possible.

In this paper we restrict our focus to atmospheric satellite sounding systems. Satellite sounding systems retrieve atmospheric variables typically as vertical profiles from top of atmosphere radiance measurements. Here we focus our attention

on modern-era hyperspectral infrared (IR) systems, and more specifically the Community Long-term Infrared Microwave Combined Atmospheric Product System (CLIMCAPS; Smith and Barnet, 2019), which is the National Aeronautics and Space Administration's (NASA) system for sounder instruments on the polar-orbiting satellites, Aqua (2002–present), Suomi-NPP (2012–present) and NOAA20 (2017–present), which is the first of the Joint Polar Satellite System (JPSS) series of four satellites scheduled to maintain operational orbit throughout 2040. CLIMCAPS implements Bayesian Optimal Estimation

(OE) (Rodgers, 2000) as inversion technique and employs explicit background error quantification with uncertainty propagation. Other sounding systems offer variations of the OE approach in practice, depending on their respective data product requirements (Susskind et al., 2003, 2014; Fu et al., 2016; DeSouza-Machado et al., 2018; Irion et al., 2018). We designed CLIMCAPS to achieve and maintain consistent observing capability across different satellite platforms so that we can generate a long-term, continuous record of satellite soundings for a nearly two-decade period of hyperspectral IR

observations from space.

Smith and Barnet (2019) described how CLIMCAPS quantifies and propagates scene-dependent uncertainty using error covariance matrices (ECM) in a sequential retrieval approach that starts with retrieving clouds, followed by temperature, water vapor and trace gas species, $O_3$, $CO$, $CH_4$, $CO_2$, $N_2O$, $SO_2$ and $HNO_3$. Averaging kernel matrices (AKMs) characterize the degree to which each of the retrieved variables depends on information contributed by the measurements about the true state

of that variable. Averaging kernels have value in data inter-comparison studies (Rodgers and Connor, 2003; Maddy and Barnet, 2008; Maddy et al., 2009; Gaudel et al., 2018; Iturbide-Sanchez et al., 2018) and form a critical component of data assimilation models (Levelt et al., 1998; Clerbaux et al., 2001; Yudin, 2004; Segers et al., 2005; Pierce et al., 2009; Liu et al., 2012).



We direct our efforts in this paper to characterizing CLIMCAPS Version 2.0 AKMs for a range of different retrieval variables, different scenes across time and space as well as multiple satellite platforms and instrument types. Our goal is to

characterize CLIMCAPS observing capability to promote a better understanding of its retrieved soundings and their value in applications.

## 1.1 Terminology and notation

We define an *observing system*, such as CLIMCAPS, as the space-based instrument along with its inversion algorithm. Observing system characteristics that affect product quality include spectral resolution, spatial footprint ('pixel' or 'field of

view') size, shape and arrangement, the instrument noise, view angles across satellite swath, which for CrIS is 2,200 km (± 50 degrees), as well as effects due to the regularization and stabilization of its retrieval algorithm. With *observing system capability,* we mean the potential a space-based system has for measuring the atmospheric state at a specific scene, given the instrument type, retrieval system design and prevailing conditions. Observing capability is akin to signal to noise ratio (SNR) and should ideally be high enough to add independent, new information to background knowledge about the atmospheric state

at any given point in time and space. CLIMCAPS employs Bayesian inversion as retrieval scheme and generate AKMs to quantify the sensitivity of retrieved variables to the true state of those variables (Rodgers, 2000) as a metric of uncertainty. CLIMCAPS product files available through the NASA Earth Observing System Data Information System (EOSDIS; Ramapriyan et al., 2010) contain AKMs for seven retrieval variables – temperature (T), water vapor ($H_2O$), ozone ($O_3$), carbon monoxide (CO), methane ($CH_4$), carbon dioxide ($CO_2$) and Nitric Acid ($HNO_3$). – at every scene. We define a CLIMCAPS

retrieval *scene* (or 'field of regard') as the spatial and spectral aggregate of radiance measurements that results from performing cloud clearing (Chahine, 1982; Susskind et al., 1998; Smith and Barnet, 2019). Cloud clearing removes the radiative effect of clouds from IR measurements by aggregating cloud sensitive channels from nine neighbouring CrIS (or AIRS) instrument footprints. Cloud clearing requires no prior knowledge of scene-specific cloud properties nor does it depend on radiative transfer calculations through clouds. Instead, cloud clearing is a robust linear method that uses the 3 x 3 spatial cluster of

instrument footprints as spectrally-independent information about scene cloudiness, and together with knowledge of the cloud-free state retrieved from coincident microwave measurements (ATMS or AMSU), derives a set of cloud-cleared spectral channels for use in subsequent retrievals. In the case where no clouds are detected, the relevant channels are simply averaged across the 3 x 3 array (9 footprints in total) with the assumption that it is a uniformly clear scene. While CLIMCAPS aggregates spectral radiance before retrieval (known as an 'average-then-retrieve' approach), the retrieved soundings are still considered

instantaneous observations because CLIMCAPS limits its radiance aggregation to small spatial clusters (an aggregate scene of 3 x 3 CrIS footprints has ~50 km diameter at nadir and ~150 km at edge of scan) and performs no temporal averaging ahead of inversion. We use the term *measurement* to refer to the measured spectrum (i.e., top of atmosphere radiance either for a single footprint or cloud-cleared scene) and distinguish it from *retrieval* that is the inverse measurement, or retrieved pressure-dependent atmospheric variable at every scene (e.g. water vapor). We maintain consistency with the mathematical notations

adopted by Rodgers (2000) for the sake of simplicity and relevance to other OE systems (Bowman et al., 2006; Ceccherini et





al., 2009; Ceccherini and Ridolfi, 2010; Fu et al., 2016; DeSouza-Machado et al., 2018; Irion et al., 2018); a measured spectrum is represented by the vector $\mathbf{y}$ with $m$ spectral channels and retrieved parameter by vector $\mathbf{x}$ with $n$ vertical pressure layers (for trace gases) or $n$ pressure levels (for temperature).

This paper starts with Section 2 as an overview of the CLIMCAPS version 2 (V2) observing system and a discussion of how its OE implementation deviates from the Rodgers (2000) theoretical OE approach. We give a detailed explanation of CLIMCAPS AKMs and how they can be employed as uncertainty metrics and indicators of observing capability. In Section 3 we present CLIMCAPS AKMs for its seven retrieval variables, T, $H_2O$, $O_3$, CO, $CH_4$, $CO_2$ and $HNO_3$. We diagnose and interpret these AKMs to conclude in Section 4 with a preliminary assessment of the CLIMCAPS observing capability and the degree of continuity in its sounding observations across satellite platforms.

## 2 Data and methods

### 2.1 CLIMCAPS observing system

CLIMCAPS is NASA's sounding observing system for the Atmospheric Infrared Sounder (AIRS; Aumann et al., 2003; Chahine et al., 2006) and the Cross-track Infrared Sounder (CrIS; Han et al., 2013; Strow et al., 2013). AIRS has been on Aqua since 2002 together with the Advanced Microwave Sounding Unit (AMSU). CrIS and the Advanced Technology Microwave Sounder (ATMS) have been on Suomi-National Polar-orbiting Partnership (SNPP) since 2011 and the National Oceanic and Atmospheric Administration (NOAA20) since 2017. We gave a detailed tabulation of the main instrument characteristics in Table 1 from Smith and Barnet (2009). Hereafter we respectively refer to these various systems as CLIMCAPS-Aqua, CLIMCAPS-SNPP and CLIMCAPS-NOAA20. Traditionally, observing systems were optimized for a specific instrument suite on a target satellite platform (Susskind et al., 2003). With CLIMCAPS, we instead focus our efforts on promoting continuity in observing capability across different instrument suites and satellite platforms so that a long-term record of satellite soundings can be generated. This means we optimize our algorithm design for consistency.

AIRS and CrIS are both new-generation hyperspectral infrared sounders that measure energy emitted at the top of Earth's atmosphere in hundreds of narrow spectral channels. With such a high spectral resolution, these instruments can measure atmospheric conditions at multiple pressure layers so that vertical structure (e.g., temperature inversions and dry layers) and atmospheric composition (e.g., stratospheric $O_3$ or mid-tropospheric CO) can be retrieved and characterized. Using the principles of information theory (Shannon, 1948), Rodgers (2000) developed a method for quantifying the information content of a spectral measurement either as the number of significant eigenvectors ($k$) from a radiance decomposition, or as degrees of freedom for signal (DOF) calculated as the trace of the AKM diagonal vector. These information content metrics, DOF and magnitude of $k$, reflect the number of independent pieces of information about the vertical atmospheric state. We can calculate these metrics for simulated spectra to quantify instrument observing capability in general, given certain design criteria like spectral resolution and noise. Or we can calculate them for real spectral measurements to quantify a satellite system's observing capability under specific conditions, uncertainty regimes and retrieval algorithm design criteria.



In Figure 1a below, we depict the total information content for all spectral channels from a global ensemble of simulated AIRS and CrIS measurements, respectively. We contrast their information content with that from the European IASI

instrument (Siméoni et al., 1997; Aires et al., 2002; Chalon et al., 2017), in polar orbit on the MetOp series since 2006. Despite instrument differences such as spectral resolution, number of channels, instrument calibration and noise (Figure 1b), CrIS, IASI and AIRS all have a total information content of $k = 100$ significant eigenvectors. This means that on a global scale, all three instruments have the ability to distinguish on the order of ~100 individual Earth system variables about the vertical atmospheric state. These include thermodynamic variables, such as temperature and moisture, along multiple layers from

surface to top of atmosphere, trace gas species, cloud surface parameters.

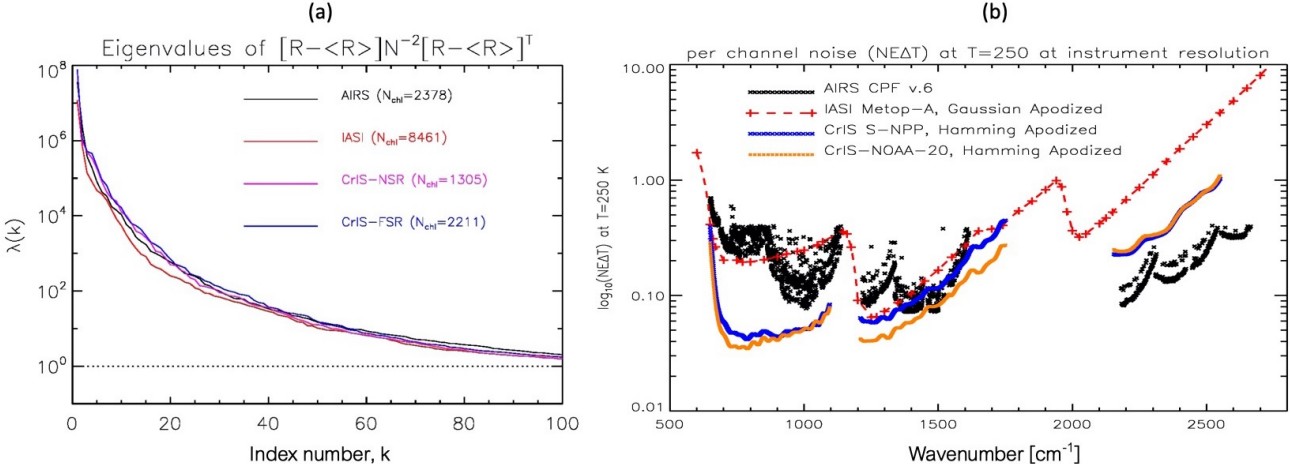

**Figure 1: Information content analysis of four operational hyperspectral infrared instruments, AIRS (Atmospheric Infrared Sounder) in orbit on Aqua since 2002, IASI (Infrared Atmospheric Sounding Interferometer) in orbit on multiple MetOp platforms since 2006 and lastly CrIS (Cross-track Infrared Sounder) in orbit on SNPP since 2011 and NOAA20 since 2017. CrIS on SNPP was**

**initially available only in Nominal Spectral Resolution (NSR) where the spectral resolution in the mid- and shortwave bands was reduced to 1.25 cm-1 and 2.5 cm-1, respectively. In 2016, CrIS on SNPP was restored to its Full Spectral Resolution (FSR) with all spectral bands sampled at 0.625 cm-1. (a) Eigenvector decomposition of the radiance covariance matrix as a measure of information content in each instrument. The eigenvalues, λ, from an eigenvector decomposition of simulated radiances are plotted against index number of each eigenvector, k. Information content is calculated as all eigenvalues λ > 0 The total number of channels, Nch, are**

**listed in the figure legend. (b) Instrument noise, measured as the noise equivalent delta temperature, NEΔT, for a scene with surface temperature equal to 250 K.**

CLIMCAPS adopted the AIRS Science Team version 5 (V5) algorithm as its baseline retrieval method, which follows a sequential OE approach in solving the non-linear inversion of infrared radiances into multiple distinct atmospheric variables. The inversion of top of atmosphere radiances is an ill-conditioned, under-determined, non-linear problem that requires some

form of stabilization to find a solution. In Bayesian (or probabilistic) OE systems, this is predominantly achieved with the introduction of an a-priori (or background) estimate of the atmospheric state, such that the solution is not an independent observation but instead represents an improvement on the background state given the top of atmosphere measurement of the true state (Rodgers, 1976, 1998, 2000).



The older AIRS V5 system employed a linear regression as a-priori for T, $H_2O$ and $O_3$ to the OE inversion step, which is
generally referred to as a 'physical' retrieval because it requires radiative transfer calculations, not regression correlation
coefficients, to minimize the cost function at every scene. CLIMCAPS does not calculate a regression a-priori T, $H_2O$ and $O_3$
but instead employs a data assimilation product; specifically the Modern-Era Retrospective Analysis for Research and
Applications Version 2.0 (MERRA2; Gelaro et al., 2017; Molod et al., 2015). We argued in Smith and Barnet (2019) that a
linear regression a-priori amplifies instrument effects in the OE retrieval and thus hamper data continuity across platforms.
Linear regression typically employs all spectral channels (Goldberg et al., 2003; Smith et al., 2012) to retrieve atmospheric
state variables simultaneously. If this regression retrieval is ingested as a-priori then instrument artefacts are propagated and
even amplified in the retrieval product because OE uses the same spectral channels (albeit a subset) a second time. CLIMCAPS
deliberately employs an instrument-independent a-priori, i.e., MERRA2, for its T, $H_2O$ and $O_3$ retrievals to minimize
instrument artefacts and promote data continuity across platforms. While MERRA2 does assimilate a small subset of IR
channels (e.g., only those sensitive to T and insensitive to $H_2O$ and other trace gases) sometimes (i.e., for clear-sky scenes
only) as part of a large suite of satellite measurements, the IR radiance information in MERRA2 is never coincident with the
retrieval scene in question because it is either absent (e.g., a cloudy scene) or from a previous orbit of measurements.
CLIMCAPS thus uses radiance information only once in the OE retrieval and any instrument artefacts that do propagate into
the retrieval null space are compensated for by MERRA2, the exact same a-priori across CLIMCAPS-Aqua, CLIMCAPS-
SNPP and CLIMCAPS-NOAA20. For the trace gas species, CO, $CH_4$, $CO_2$, $HNO_3$, $N_2O$ and $SO_2$, CLIMCAPS employs a
static global climatology.

The CLIMCAPS retrieval algorithm is outlined in Figure 2 and we highlight four major steps here. (1) *Local angle
correction* that removes satellite view angle differences among a spatial cluster of 3 x 3 instrument footprints also known as
the "field of regard" or retrieval scene. (2) *MW-only retrieval* that retrieves vertical profiles of T, $H_2O$ and liquid water path
(LIQ), as well as surface emissivity (ε) using spectral channels from the microwave measurements (AMSU on Aqua, ATMS
on SNPP and NOAA20). This results in an estimate of cloud-free vertical atmospheric structure in all but precipitating scenes.
(3) *Cloud Clearing* that removes the radiative effects of clouds from hyperspectral IR channels in each field of regard using
MW-only retrievals of LIQ and ε from Step 2, profiles of T, $H_2O$ and $O_3$ from MERRA2 as well as climatologies of CO, $CH_4$,
$CO_2$, $HNO_3$, $N_2O$ and $SO_2$. Cloud clearing is described in detail elsewhere (Smith, 1968; Chahine, 1974, 1977, 1982; Susskind
et al., 2003) and remains one of the most robust approaches for the retrieval of atmospheric parameters within complex cloudy
conditions and up to 90% cloud cover. This step aggregates the cluster of 3 x 3 IR spectra into a single cloud-cleared IR
spectrum from which all subsequent retrievals are done. In the case where a scene has no cloud cover or where IR channels
are insensitive to clouds, the 3 x 3 cluster of IR channels is simply averaged. Note that cloud clearing reduces the spatial
resolution of CrIS or AIRS footprints from ~15 km instrument resolution at nadir to ~50 km at nadir. (4) *Final retrieval* that
sequentially retrieves surface temperature ($T_s$), ε, reflectivity (ρ), T, $H_2O$ and $O_3$, CO, $CH_4$, $CO_2$, $HNO_3$, $N_2O$ and $SO_2$. It is
important to note that for cloud-cleared scenes, the profile retrievals do *not* represent conditions within the cloud fields but
rather around the clouds. This is a subtle distinction, but meaningful in scientific studies and applications.


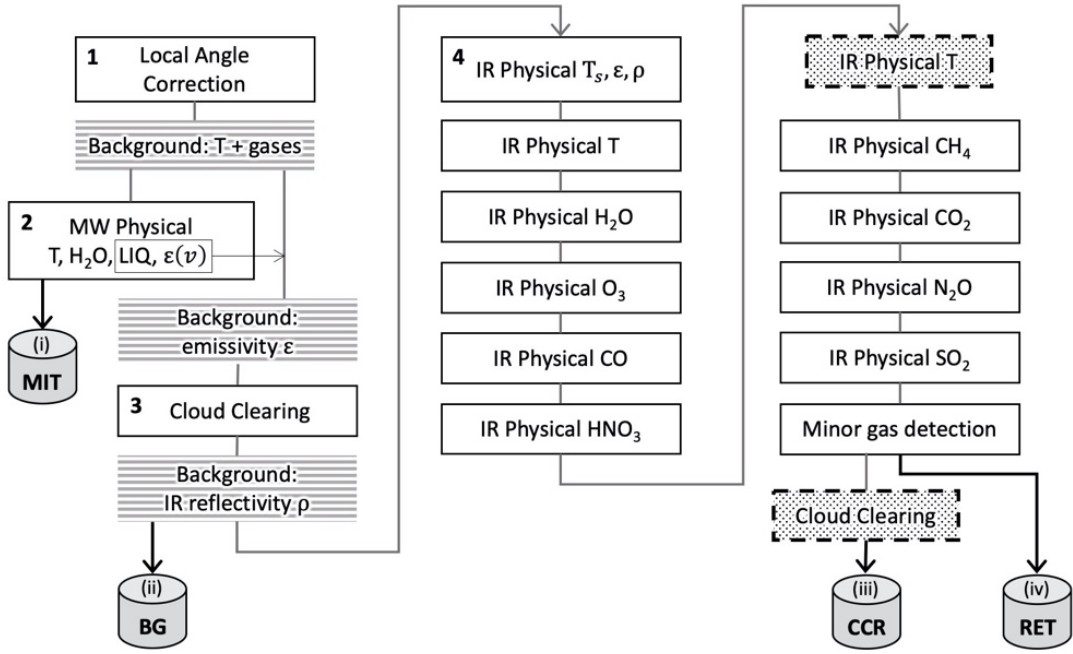

**Figure 2: High-level abstract representation of the CLIMCAPS retrieval algorithm highlighting four categories of contributions to the final product file, (i) the microwave-only retrieval parameters (MIT), (ii) the background (BG) estimates for temperature, T(p), moisture, H₂O and ozone O₃ from the MERRA2 reanalysis product, surface emissivity over land (CAMEL database) and ocean (Masuda model) as well as infrared reflectivity, (iii) cloud cleared radiances (CCR) calculated from the final set of retrieved parameters, and (iv) the retrieved atmospheric parameters (RET) for T, H₂O and all the minor gases. Retrieval steps 1 through 4 are discussed in the main text.**

Each retrieval step (Figure 2) is performed on a subset of channels with maximum sensitivity to the target variable, and minimum sensitivity to all other variables. We adopted the channel selection method as described in Gambacorta and Barnet (2013). The channel sets for cloud clearing and all trace gases – $O_3$, $CO$, $CH_4$, $CO_2$, $HNO_3$, $N_2O$ and $SO_2$ – are selected from the IR measurements only, while the channel sets for surface parameters as well as atmospheric T and $H_2O$ are selected from the IR and microwave measurements. The number of IR channels for each variable and each instrument is listed in Table 1 below and represents the size, $m$, of measurement vector, $\mathbf{y}$, for each retrieval variable. While $m$ varies among instruments and retrieval variables, the size, $n$, of retrieval vector, $\mathbf{x}$, remains constant at 100 vertical pressure levels (for temperature) and layers (for trace gas column densities) for the sake of accurate radiative transfer calculations. CLIMCAPS employs the Stand Alone Radiative Transfer Algorithm (Strow et al., 2003) originally developed for AIRS and later adopted for CrIS. Table 1 additionally lists two values; the maximum value ($B_{max}$) for each retrieval damping factor (i.e., a static scalar threshold below which spectral channels are damped according to their information content) and, the degrees of freedom for signal (DOF) as the global average of CLIMCAPS cloud-cleared radiance spectra with $m$ channels. We discuss the damping factor in Section 2.2 below, but in short, it determines the degree to which CLIMCAPS retains information from the radiance channels in the retrieved product.





**Table 1: For each CLIMCAPS instrument/platform configuration, we list three parameters, the number of spectral channels (nch) used in the retrieval of temperature, H₂O, O₃, CO, CH₄, CO₂ and HNO₃, the damping factors applied as regularization parameter ($B_{max}$) and degrees of freedom as metric for vertically integrated observing capability. CLIMCAPS version 2.0 is configured for retrievals from (i) Atmospheric Infrared Sounder (AIRS) on Aqua, (ii) Cross-track Infrared Sounder in Nominal Spectral Resolution mode (CrIS-NSR) on Suomi National Polar orbiting Partnership (SNPP), (iii) CrIS in Full Spectral Resolution mode**
**(CrIS-FSR) on SNPP, and (iv) CrIS-FSR on NOAA20, the first of four Joint Polar Satellite Systems.**

|  | (i) Aqua/AIRS | | | (ii) SNPP/CrIS-NSR | | | (iii) SNPP/CrIS-FSR | | | (iv) NOAA20/CrIS FSR | | |
|---|---|---|---|---|---|---|---|---|---|---|---|---|
|  | nch | $B_{max}$ | DOF | nch | $B_{max}$ | DOF | nch | $B_{max}$ | DOF | nch | $B_{max}$ | DOF |
| Temperature | 134 | 0.25 | 5.9 | 86 | 0.2 | 3.4 | 120 | 0.2 | 2.9 | 120 | 0.2 | 3.0 |
| Water Vapor (H₂O) | 46 | 0.4 | 2.4 | 62 | 0.4 | 2.1 | 66 | 0.4 | 1.6 | 66 | 0.4 | 1.6 |
| Ozone (O₃) | 40 | 1.0 | 1.9 | 53 | 1.0 | 2.3 | 77 | 1.0 | 1.9 | 77 | 1.0 | 1.8 |
| Carbon Monoxide (CO) | 36 | 1.85 | 0.6 | 27 | 1.85 | 0.2 | 35 | 1.85 | 0.8 | 35 | 1.85 | 0.8 |
| Methane (CH₄) | 65 | 1.25 | 0.9 | 55 | 1.25 | 0.6 | 84 | 1.25 | 0.7 | 84 | 1.25 | 0.7 |
| Carbon Dioxide (CO₂) | 61 | 0.38 | 0.5 | 53 | 0.38 | 0.9 | 54 | 0.38 | 0.8 | 54 | 0.28 | 0.8 |
| Nitric Acid (HNO₃) | 14 | 1.0 | 0.3 | 28 | 1.0 | 0.3 | 30 | 1.0 | 0.1 | 30 | 1.0 | 0.1 |

## 2.2 CLIMCAPS averaging kernels

Rodgers (2000) defines averaging kernels as the sensitivity of the retrieved parameter, $\hat{x}$, to the true state of the parameter, $x$, for a given moment in time and space. In its most basic form, an $n \times n$ AKM can be calculated for each retrieved parameter as depicted in Eq. (1):

$$AKM = \left[K^T S_m^{-1} K + S_a^{-1}\right]^{-1} K^T S_m^{-1} K, \tag{1}$$

where $K$ is the $m \times n$ matrix of weighting functions (or Jacobians) that characterizes measurement sensitivity to the a-priori target variable as $\frac{\partial y}{\partial x_a}$, $S_m$ is a diagonal $m \times m$ matrix of instrument noise and $S_a^{-1}$ the regularization term, which in Rodgers' (2000) approach is defined by the inverse of an $n \times n$ a-priori error covariance matrix, $S_a$. The value of $S_a$ determines the amount of regularization applied to the retrieval step, or the degree to which information content in the spectral measurement

contributes to the final result. $S_a$ has to be chosen carefully so that the information content of the retrieval (or regularized solution) can be optimized given the information content available in the measurement (von Clarmann and Grabowski, 2007). In a Bayesian OE system, the regularization term determines how much the retrieved variable resembles the a-priori variable. If $S_a$ is low, then regularization is high and the measurement information content will be suppressed so that the retrieval more closely resembles the a-priori. In most OE observing systems, it is computationally prohibitive to dynamically generate a

scene-specific matrix, $S_a$, especially where data latency is a concern. Instead, a common approach is to set $S_a$ to a static value that is calculated offline either as a statistical covariance of a data ensemble or a simple ad hoc assignment (Fu et al., 2016; Irion et al., 2018). $S_a$ is then applied to each retrieval scene irrespective of the measurement information content for that scene.



While this simplifies calculation, it risks suppressing information content where it is high, or enhancing measurement uncertainty where information content is low. The Rodgers (2000) AKM (Eq. 1) can be described as a linear combination of
measurement sensitivity weighted by uncertainty about the priori state variable ($\mathbf{S}_a$).

CLIMCAPS, in contrast, calculates an $n \times n$ AKM as in Eq (2):

$$\mathbf{AKM} = [\mathbf{K}^{\mathbf{T}}\mathbf{S}_m^{-1}\mathbf{K} + \lambda]^{-1}\mathbf{K}^{\mathbf{T}}\mathbf{S}_m^{-1}\mathbf{K} \tag{2}$$

with $\mathbf{K}$ the same as in Eq. (1), but $\mathbf{S}_m$ an $m \times m$ error covariance matrix that combines instrument noise with uncertainty from scene-specific and observing system effects as described by Smith and Barnet (2019). Moreover, the background error term,
$\mathbf{S}_a$ in Eq. (1), is replaced here with $\lambda$, the damping factor listed in Table 1. This damping factor differs from $\mathbf{S}_a$ in two important ways; (i) unlike $\mathbf{S}_a$, $\lambda$ has horizontal variation because it is dynamically calculated for each retrieval scene based on the measurement information content for a target variable and, (ii) unlike $\mathbf{S}_a$, $\lambda$ has no vertical variation because it is a scalar value that assumes uniform uncertainty about the prior state, which can be an oversimplification in some cases. In contrast to Eq. (1), a CLIMCAPS AKM as in Eq. (2) can be described as the linear combination of measurement sensitivity weighted by known
and propagated sources of uncertainty as well as scene-specific knowledge about measurement information content. While this is different from a traditional OE approach, both Eq. (1) and (2) generate results that are within the observing system null space and thus part of the solution set of the ill-determined inversion problem.

CLIMCAPS adopted the AIRS Version 6 (Susskind et al., 2003, 2014) implementation of Eq. 2 (Maddy et al., 2009; Maddy and Barnet, 2008). Instead of an array size of $n = 100$, CLIMCAPS calculates AKMs on a reduced set of pressure
layers as defined by a series of overlapping trapezoidal functions. The thickness of each trapezoid layer is empirically determined from calculations of the vertical resolution of simulated measurements for each parameter, e.g. CLIMAPS has 31 trapezoids for temperature and 9 for CO. These trapezoids were selected by the AIRS V5 science team, with ~2 trapezoids per retrievable layer quantity. CLIMCAPS employs these vertical trapezoidal functions for a number of reasons; (i) they reduce the dimensionality of the Jacobian matrix to speed up algorithm processing time; (ii) compared to the 100 pressure layers
needed for accurate radiative transfer calculation, the trapezoidal layers more closely resemble true instrument vertical resolution calculated from simulated spectra for standard atmospheric state climatologies; and (iii) they act as smoothing constraint by reducing the need for additional a-priori terms to stabilize the solution. As mentioned, we use Rodgers' (2000) OE notation in this paper, but in practice the Jacobians in Eq. (2) are linearly transformed to the coarser trapezoidal grids using a transformation matrix $\mathbf{W}$ as follows: $\widetilde{\mathbf{K}} = \mathbf{KW}$ making it a $m \times \tilde{n}$ matrix with $\tilde{n}$ the number of trapezoid layers (see Maddy
and Barnet 2008 for more details).

Averaging kernels are unitless and typically range in value between 0.0 and 1.0, although they can sometimes have negative values where noise exceeds signal (see Figure 3 in Section 3 below). AKMs quantify CLIMCAPS observing capability at any given point in time and space because they account for all known sources of scene-specific as well as observing system uncertainty. They characterize a system's ability for observing a target parameter at a specific scene. An
alternative interpretation is that they quantify the degree to which the a-priori variable compensates for the lack of observing





capability at any specified scene (1.0 – AKM). While CLIMCAPS AKMs do not measure retrieval accuracy (approximation to the truth), they do characterize retrieval uncertainty and information content. CLIMCAPS retrievals are not in situ measurements of the vertical atmospheric state, but under-determined non-linear inverse measurements with dependence on prior knowledge of the atmospheric state. In scientific analysis and operational applications, it is imperative that sounding

observations are correctly interpreted least their uncertainty is mistaken for measurements. CLIMCAPS AKMs characterize and quantify, in a Bayesian sense, the weighted contributions from the measurements (0.0 + AKM) as well as the a-priori (1.0 – AKM). An averaging kernel value of zero means that the measurement has no observing capability at that pressure layer and the solution will be the a-priori. An averaging kernel value of unity means the measurement has 100% observing capability and the solution will have no dependence on the a-priori. In practice, however, averaging kernels range in value between these

two endpoints such that $0.0 < AKM < 1.0$.

What can we learn about CLIMCAPS observing capability by diagnosing its AKMs? And how should we interpret differences between its retrievals from different parts of the globe or from different sounding systems? We can address these questions with a discussion of how each of the variables in Eq. (2) affect the AKMs. These are the Jacobians (**K**) that determine the structure of an AKM, and the measurement error covariance matrix (**S**$_m$) with regularization parameter ($\lambda$) that determine

its magnitude.

CLIMCAPS Jacobians are finite differencing (or brute force) weighting functions that quantify the sensitivity of the calculated radiances to the a-priori retrieval variable. They are $m \times n$ matrices, with $m$ equal to the number of spectral channels in the retrieval subset (Table 1); out of 2211 CrIS channels, CLIMCAPS has $m = 120$ selected for T and $m = 66$ for $H_2O$. Jacobians are sensitive to the background state variables used in the forward radiative transfer calculation. This is the only

parameter in Eq. (2) that ingests a-priori information. If an a-priori is biased with respect to the true background state, the same bias will propagate into the Jacobians. For example, if the CO a-priori is a climatology of a typical source site, then the Jacobian will indicate high measurement sensitivity because high concentrations of mid-tropospheric CO result in strong absorption lines in the calculated radiance, and thus yield large weighting functions. If such weighting functions are applied to a retrieval where the scene-specific CO concentrations are low, then the averaging kernels will mistakenly indicate high observing

capability to CO at that scene, which risks enhancing the uncertainty as signal. That is, unless the averaging kernels are adjusted according to known sources of uncertainty.

Clouds introduce one of the primary sources of scene-specific uncertainty. While CLIMCAPS requires no knowledge about the a-priori state of clouds, it calculates radiance uncertainty due to clouds in the cloud clearing step (Table 1). This, together with uncertainty from other retrieval variables are propagated into the measurement error covariance matrix, **S**$_m$,

according to the method described in Smith and Barnet (2019). If a scene has high uncertainty due to clouds, **S**$_m$ will increase and AKM will decrease to reflect a reduced observing capability. Scene-dependent cloud effects are therefore not explicitly accounted for in AKMs through radiative transfer calculation, but their scene-dependent uncertainty is derived and propagated into one of the error terms.


CLIMCAPS performs singular value decomposition (SVD) of $\widetilde{\mathbf{K}}^T \mathbf{S}_m^{-1} \widetilde{\mathbf{K}}$ to derive a set of scene-specific eigenvectors for
use in the retrieval. We refer to this $\tilde{n} \times \tilde{n}$ eigenvector matrix as $\overline{\overline{\mathbf{K}}}$, with eigenvalues, $\lambda_i$, on its diagonal. SVD benefits the
retrieval in that it minimizes (maximizes) the a-priori contribution when measurement information content is high (low) such
that the retrieval product deviates from its a-priori only when the radiance measurement has information content. According
to Eq. (2), the regularization term is derived from the eigenvalues and determines the degree to which these eigenvectors are
damped in the solution according to the critical threshold, $\lambda_c$, which is derived from $B_{max}$ (Table 1) such that $\lambda_c = (B_{max})^{-2}$.
$B_{max}$ is a scalar value, empirically determined offline, and defines the maximum allowable noise that can propagate into the
retrieval. We illustrate how this works in practice with the example discussed below.

**Table 2: Example of eigenvalues and damping factors for a hypothetical temperature retrieval.**

| | $B_{max} = 0.5 \rightarrow \lambda_c = 4.0$ | | | |
|---|---|---|---|---|
| $i$ | $\lambda_i$ | $\Delta\lambda$ | Percent damped | |
| 1 | 18.719 | 0.0 | 0.0 % | Not damped |
| 2 | 8.321 | 0.0 | 0.0 % | Not damped |
| 3 | 4.934 | 0.0 | 0.0 % | Not damped |
| 4 | 3.127 | 0.41 | 11.58 % | Damped |
| 5 | 1.312 | 0.98 | 42.73 % | Damped |
| 6 | 0.68 | 0.97 | 58.77 % | Damped |
| 7 | 0.29 | 0.79 | 73.07 % | Damped |
| … | … | … | … | … |
| 22 | 1.4e-07 | 4.1e-04 | 100.0 % | Switched off |
| 23 | 4.3e-08 | 2.0d-04 | 100.0 % | Switched off |

In Table 2, the $\overline{\overline{\mathbf{K}}}$-matrix for temperature has five significant eigenvalues (i.e., where $\lambda_i \geq 1.0$), which means that the
observing system has five independent pieces of information and can solve for temperature at five distinct pressure levels. For
a $B_{max} = 0.5, \lambda = 4.0$. All eigenvectors with $\lambda_i > \lambda_c$ will contribute to the retrieval undamped. In Table 1, we see that the
first three eigenvectors will thus contribute 100% of their information to the retrieval. Those eigenvectors with $\lambda_c > \lambda_i > 0.05$ will be fractionally damped as follows: $1.0 - \frac{\lambda_i}{(\lambda_i + \Delta\lambda)}$, where $\Delta\lambda = \sqrt{\lambda_c}\sqrt{\lambda_i} - \lambda_i$. Accordingly, the fourth eigenvector
(Table 2) will be 11.58% damped, the fifth 42.73% and so on. Those eigenvectors with $\lambda_i < 0.05$ will be switched off so that
they make no contribution to the retrieval because they are regarded as sources of noise. An observing system can be over-
damped in which case it does not let enough functions contribute 100% of their information. Such a system would suppress
the amount of information contributed by the measurements and force a strong dependence on the a-priori. Alternatively, a
system can also be under-damped in which case too many functions contribute to the retrieval undamped such that the
measurements contribute not only information (eigenvectors with $\lambda_i \geq 1.0$) but also noise (eigenvectors with $\lambda_i < 1.0$).



CLIMCAPS-Aqua has $B_{max} = 0.25$ and CLIMCAPS-NOAA20 $B_{max} = 0.2$ (Table 1), which translates to $\lambda_c = 16.0$ and $\lambda_c = 25.0$, respectively. In our example given in Table 2, CLIMCAPS-Aqua will leave only the first eigenvector undamped, while CLIMCAPS-NOAA20 will not let a single eigenvector contribute 100% of its information, but damp all of them.

    We adopt this type of regularization in CLIMCAPS because we do not know with absolute certainty that we fully accounted for all sources of uncertainty in the $\mathbf{S}_m$ matrix. With this approach, we can account for those sources of uncertainty
not explicitly characterized in previous retrieval steps (Figure 1). In an ideal system where all sources of uncertainty are fully characterized, all eigenvectors with $\lambda_i \geq 1.0$ will contribute to the retrieval undamped.

## 3 Results and discussion

In this section, we use AKMs to diagnose CLIMCAPS observing capability (or sensitivity to the true state) for CLIMCAPS-Aqua and CLIMCAPS-NOAA20 using two global days of retrievals, 1 July and 15 December 2018. AKMs quantify the
potential each measurement has to resolve the atmospheric state, given observing system characteristics and prevailing conditions at the retrieval scene. So far, we referred to the AKM associated with each retrieval. Here we take a look at the individual averaging kernels (or rows) of each AKM and specifically distinguish the diagonal of the AKM (or AKD) as a vector representation of the maximum sensitivity at each pressure level.

### 3.1 Diagnosing CLIMCAPS observing capability

Figure 3 below depicts the averaging kernels for T and $H_2O$ from CLIMCAPS-NOAA20 for five different retrieval scenes within a few hundred miles of each other south of South Africa where the Atlantic and Indian Oceans converge. The peak of each kernel depicts the atmospheric pressure level where observing capability is strongest. The spread of an averaging kernel, quantified as the full-width at half-maximum (FWHM), can be interpreted as the vertical resolution of information content at its peak pressure. Accordingly, we see here that CLIMCAPS has higher vertical resolution (smaller FWHM) for T in the lower
troposphere (Figure3; top row), but stronger observing capability in the stratosphere (larger peak values) but with lower vertical resolution (larger FWHM). The vertical resolution for $H_2O$ (Figure 3; bottom row) is fairly consistent throughout the troposphere but we see how observing capability varies strongly from scene-to-scene. Note how the kernels fall below zero at times. For scene 1 and 3 ([-47.8, 29.4] and [-41.7, 22.6], respectively) the kernels for both T and $H_2O$ are generally low in the troposphere compared to other scenes. This means the observing capability of CLIMCAPS-NOAA20 is weak and only a small
amount of measured information will be added to the a-priori at those scenes. Scene 4 ([-36.6, 29.9]), on the other hand, has higher kernel peaks and CLIMCAPS-NOAA20 thus has a stronger capability to retrieve atmospheric structure in the troposphere and add new information to prior state variables at that scene.

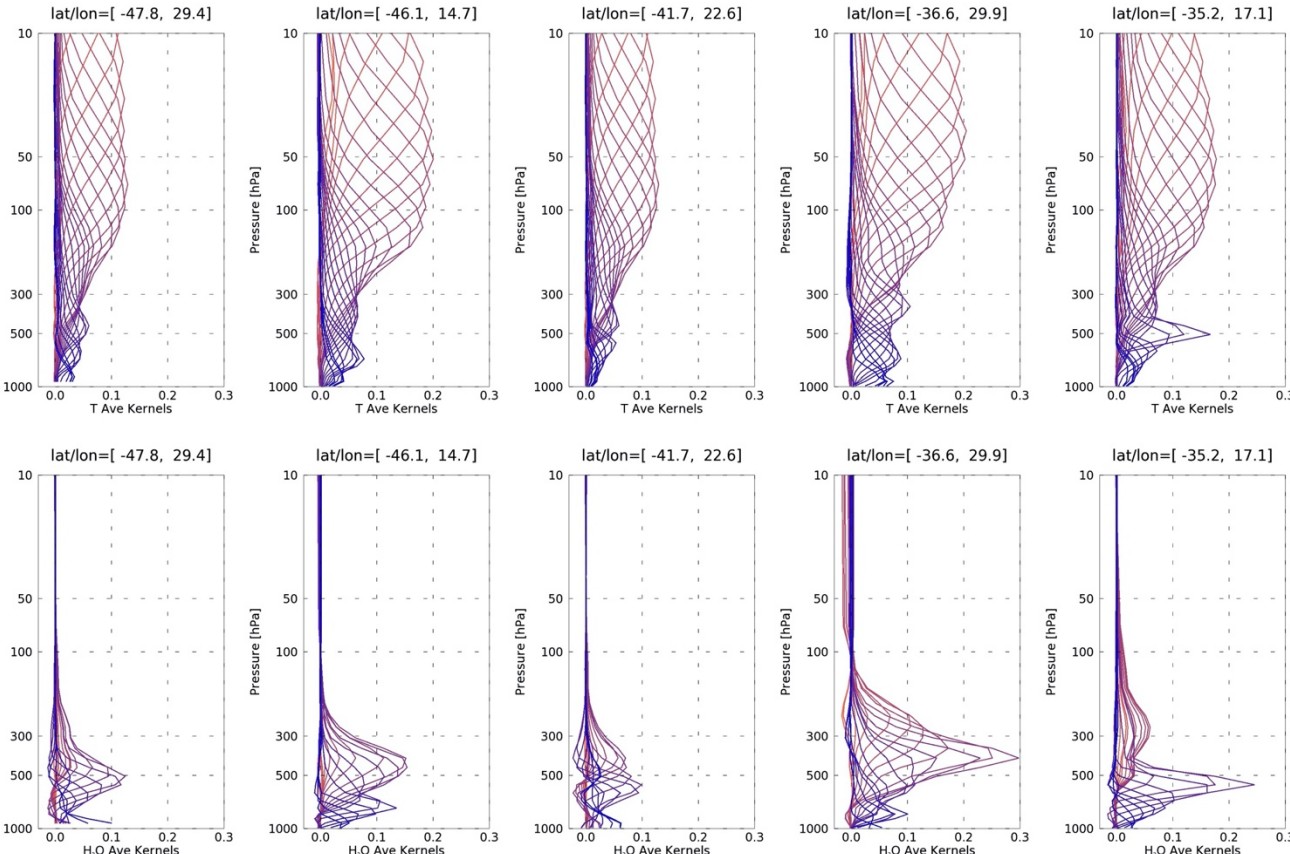

**Figure 3: Scene-dependence of CLIMCAPS-NOAA20 averaging kernels for coincident (top row) temperature (T) and (bottom row)**
**water vapor ($H_2O$) retrievals at five scenes (left to right) on 1 July 2018. The latitude/longitude coordinates are listed at the top of**
**each figure. Averaging kernels (Eq. 2) quantify and characterize the signal-to-noise ratio of an observing system and are affected by**
**the scene-dependent effects (e.g., temperature lapse rate, amount of gas molecules, surface emissivity and cloud uncertainty) as much**
**as the measurement characteristics (e.g., spectral resolution, instrument calibration and noise). CLIMCAPS retrieves T and $H_2O$**
**sequentially each with a unique subset of channels, which means that the variation in these averaging kernels are independent of**
**each other.**

Figure 4 presents the averaging kernels for seven CLIMCAPS-NOAA20 retrieval parameters. They are (left to right),

T, $H_2O$, $O_3$, CO, $CH_4$, $CO_2$ and $HNO_3$. These kernels represent the average for all Northern Mid-Latitude scenes (30˚–60˚N,

180˚W–180˚E) on 1 July 2018, hence their smooth appearance compared to those in Figure 3 for individual scenes. We see

how retrieval sensitivity to the true state depends strongly on the target variable. CLIMCAPS retrieves each state variable

using a subset of spectral channels (Table 1) selected to have a high degree of sensitivity for the target variable and low

sensitivity to all other atmospheric variables radiatively active in the same spectral region (Gambacorta and Barnet, 2013). The

CLIMCAPS sequential OE approach with channel selection and uncertainty propagation, minimizes spectral correlation in the

retrieved variables (Smith and Barnet, 2019). This means that any correlation that do exist can mostly be attributed to

geophysical, and not observing system, effects. On average, CLIMCAPS-NOAA20 has distinct stratospheric and tropospheric




sensitivity to the true states of T, $O_3$ and $CO_2$. For $H_2O$, CO and $CH_4$, CLIMCAPS-NOAA20 observing capability is limited

to the mid-troposphere (200–700 hPa). Unlike CO and $CH_4$, the kernels for $H_2O$ have peaks at multiple layers and varying

degrees of vertical resolution (FWHM). On average in the summertime Northern Mid-Latitude zone, CLIMCAPS-NOAA20

has barely any sensitivity to $HNO_3$ and very little to $CO_2$ below 500 hPa.

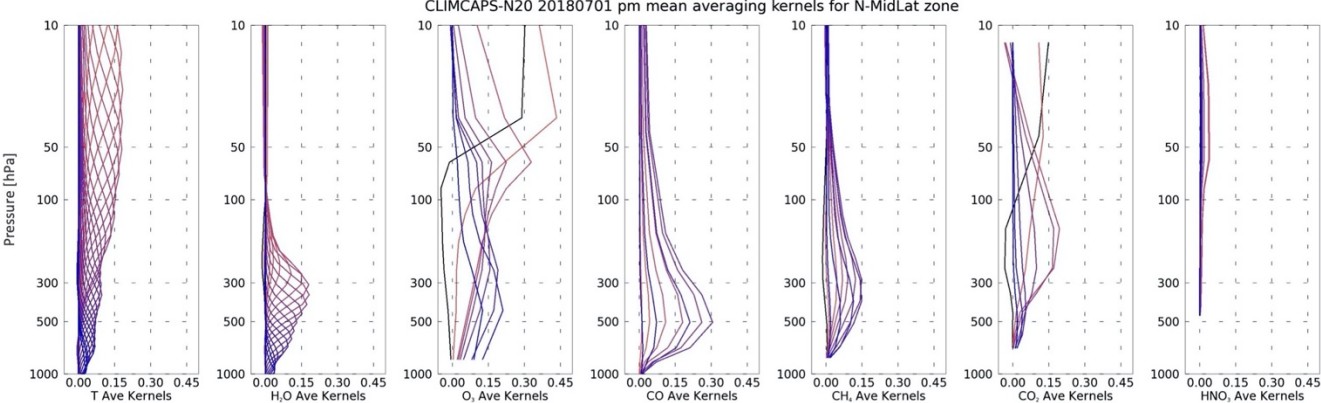

**Figure 4: The mean of a set of averaging kernels for seven CLIMCAPS-NOAA20 ascending orbit retrieval variables across the**
**North Mid-Latitude zone (30˚N to 60˚N) for a global day of daytime (ascending orbit) observations from NOAA20 on 1 July 2018.**
**From left to right is air temperature (T), water vapor ($H_2O$), ozone ($O_3$), carbon monoxide (CO), methane ($CH_4$), carbon**
**dioxide ($CO_2$) and nitric acid ($HNO_3$). CLIMCAPS calculates 31 averaging kernels for T, 22 for $H_2O$, 10 for $O_3$, CO and $HNO_3$, 11**
**for $CH_4$ and 9 for $CO_2$. The averaging kernels for T, $H_2O$ and CO are defined on layers from top of atmosphere to sea surface, with**
**those for $O_3$ extending down to 822 hPa, $CH_4$ down to 800 hPa, $CO_2$ down to 700 hPa and $HNO_3$ down to 450 hPa.**

To simplify comparison across multiple latitudinal zones and retrieval systems, we use Averaging Kernel Matrix Diagonal

vectors (in short, AKDs from here on) to summarize the maximum sensitivity at each pressure layer. The trace of the AKM

(sum of AKD) defines the "degrees of freedom for signal" (DOF), or the damped information content about the vertical state

of a target variable. DOF is can be smaller than the number of significant eigenvectors due to damping (Eq. 2) and can be

interpreted as the SNR of a retrieval system.

In Figure 5 below, we contrast the AKDs for five latitudinal zones – South Polar (90˚S to 60˚S), Southern Mid-Latitude

(60˚S to 30˚S), Tropics (30˚S to 30˚N), Northern Mid-Latitude (30˚N to 60˚N) and North Polar (60˚N to 90˚N) – on

15 December 2018 for CLIMCAPS-NOAA20 (top panel) and CLIMCAPS-Aqua (bottom panel). We observe distinct

latitudinal variation in CLIMCAPS-NOAA20 for $H_2O$, $O_3$ and $CO_2$. In contrast, CLIMCAPS-Aqua information content has

latitudinal variability for T, $H_2O$, $O_3$ and $HNO_3$. For CO and $CH_4$, CLIMCAPS-NOAA20 and CLIMCAPS-Aqua information

content is similar in magnitude and structure with mid-tropospheric peaks at 500 hPa and 400 hPa, respectively. Notice the

marked differences in T and $H_2O$ AKDs between CLIMCAPS-Aqua and -NOAA20 (Figure 5, two left panels). Compared to

CLIMCAPS-NOAA20, CLIMCAPS-Aqua has higher observing capability for atmospheric structure in the mid-troposphere;

its T and $H_2O$ retrievals have smaller dependence on the a-priori with a larger contribution of information by the AIRS/AMSU

spectral channels. Both observing systems use the same a-priori, namely MERRA2, and they measure conditions on the same

day. While Aqua and NOAA20 both have 13h30 local overpass times their orbits are, however, not aligned and they view the



same scene at different view angles almost an hour apart. Cloud structure and amount can change dramatically in that time. But even if the cloud fields remained unchanged over a few hours, measurement uncertainty due to clouds can be different at nadir (looking down at clouds) than edge of scan (looking at clouds with an angle). Smith et al. (2015) discussed how observing

capability changes due to instrument effects – spectrometers (AIRS) versus interferometers (CrIS) – in cloudy scenes. While the information content for an ensemble of simulated AIRS and CrIS measurements are similar (Figure 1), differences in their spectral resolution, detector arrays, and algorithm channel sets introduce variation in the information content of their measurements at a specific same scene. CLIMCAPS-Aqua uses 134 and 46 channels for T and $H_2O$, while CLIMCAPS-NOAA20 uses 120 and 66 for the same variables, respectively. Moreover, the damping factor for CLIMCAPS-Aqua T is lower

than that for CLIMCAPS-NOAA20.

We designed and implemented CLIMCAPS to be similar for all instruments and platforms with the goal that its sounding record can be continuous over decades despite changes in technology. Global ensembles of T and $H_2O$ retrievals from both systems – CLIMCAPS-NOAA20 and CLIMCAPS-Aqua – display similar root-mean-square statistics (not shown) when compared to ECMWF reanalysis fields (European Centre for Medium-Range Weather Forecasts). We have found that

CLIMCAPS-NOAA20 and CLIMCAPS-Aqua have similar observing capabilities for the trace gases, but compared to CLIMCAPS-Aqua, CLIMCAPS-NOAA20 appears over-damped; its T and $H_2O$ retrievals have low sensitivity to the true state. This is reflected in the CLIMCAPS regularization threshold for T from CrIS/ATMS on SNPP and NOAA20 that is lower than that for AIRS/AMSU on Aqua (Table 1). This threshold was first developed for nominal spectral resolution CrIS (measurements available at launch in 2011) and never updated when full spectral resolution CrIS measurements became

available two years later. We will re-evaluate this in future to test if the information content for these two systems can be optimized for consistency or if fundamental instrument differences will prohibit a continuity in information content.



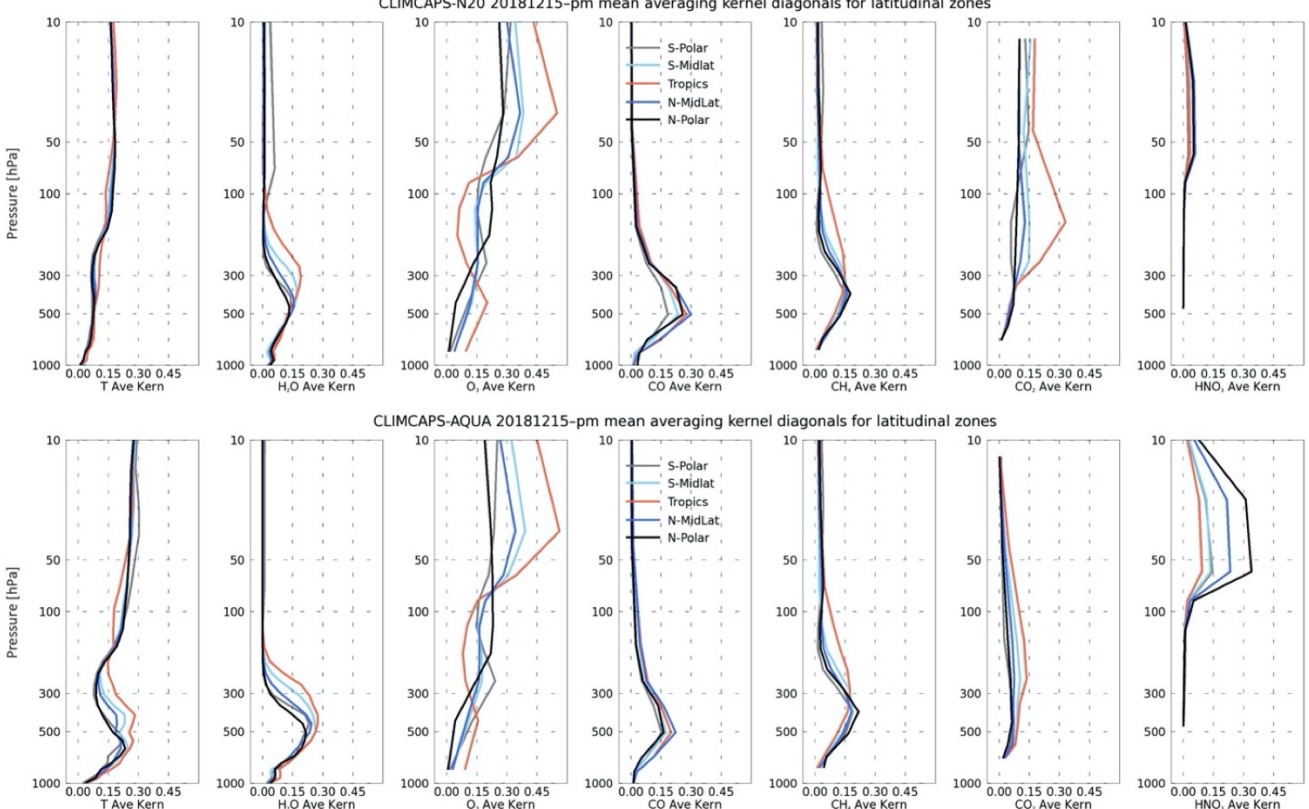

**Figure 5: Averaging kernel diagonal vectors for seven retrieval variables – (left to right) T, $H_2O$, $O_3$, CO, $CH_4$, $CO_2$ and $HNO_3$ – from (top) CLIMCAPS-NOAA20 and (bottom) CLIMCAPS-Aqua ascending orbits on 15 December 2018. For each observing system, the mean of the diagonal vector is calculated across five latitudinal zones – South Polar (90°S to 60°S), Southern Mid-Latitude (60°S to 30°S), Tropics (30°S to 30°N), Northern Mid-Latitude (30°N to 60°N) and North Polar (60°N to 90°N).**

Figure 6 maps CLIMCAPS-NOAA20 DOF for T, $H_2O$, CO and $O_3$ on 15 December 2018. CLIMCAPS AKMs are independent of the final retrieved variable and thus independent of whether the solution converges or not. We, therefore, do not apply a quality control filter that introduces data gaps other than those introduced by orbital tracks at low latitudes. Note how the spatial patterns of DOF for the four variables are largely independent of each other. This stems from the fact that CLIMCAPS uses channel subsets and uncertainty propagation to minimize spectral dependence in retrieved variables (Smith and Barnet, 2019).





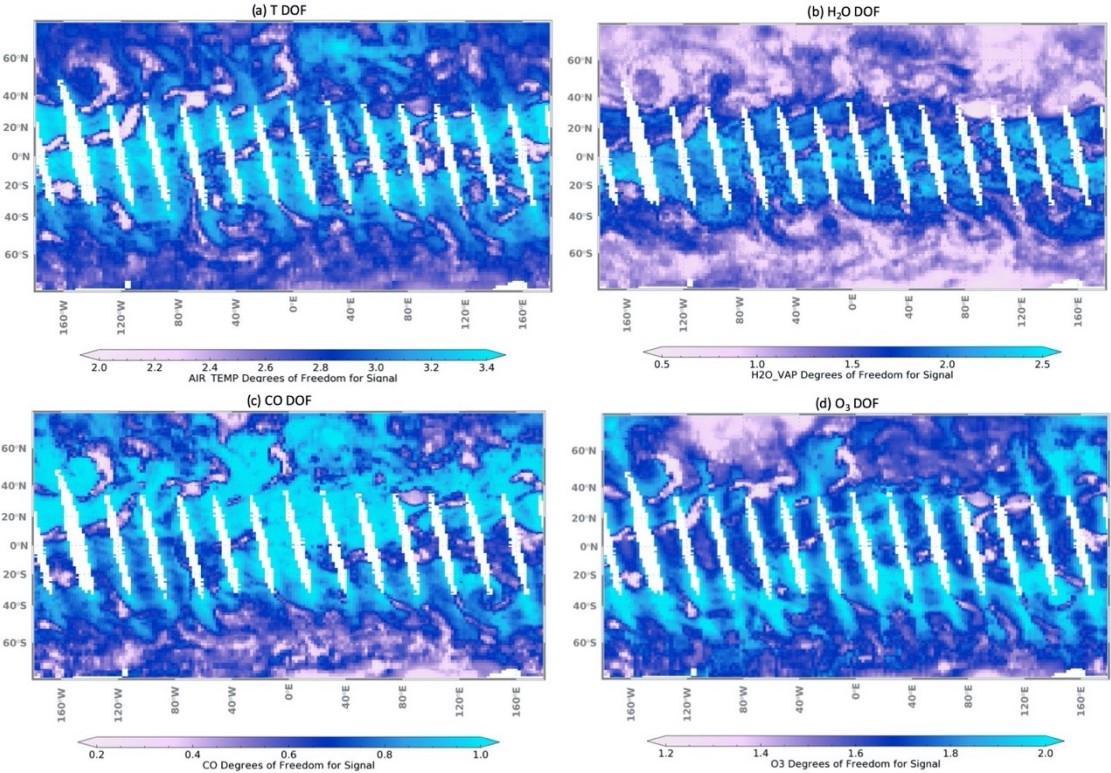

**Figure 6: Spatial variation in the degrees of freedom for signal (DOF) for four retrievals from CLIMCAPS-NOAA20 ascending**
**orbit on 15 December 2018, (a) temperature (T), (b) water vapor (H₂O), (c) carbon monoxide (CO) and (d) ozone (O₃). Note how the**
**spatial patterns in DOF for each retrieval variable is largely independent of the others.**

While CLIMCAPS observing capability for these variables are largely independent of each other, their spatial patterns do
all display a sensitivity to clouds in the lower latitudes. We see similar patterns in cloud cover from satellite imagery of the
same day (not shown). AKMs do not *directly* ingest any information about the background atmospheric state or the a-priori
retrieval variable. Neither do the AKMs ingest any cloud parameters in radiative transfer calculations for deriving the **K**-
matrix. Any knowledge about clouds that does exist in the AKMs (and derived DOF) is from the cloud uncertainty that is
quantified during the cloud clearing step and propagated through to the $S_m$ matrix. If cloud uncertainty is high, $S_m$ will increase
and DOF will decrease according to Eq. (2). This is why we see lower values for DOF in cloudy and overcast scenes.

Figure 7 illustrates the degree to which AKDs vary across a North Mid-latitude zone (30˚N to 60˚N) for seven retrieval
variables; from left to right they are T, H₂O, O₃, CO, CH₄, CO₂ and HNO₃. The solid lines represent their mean AKDs with
the error bars quantifying their variation about the mean. The degree to which the AKDs vary across space, pressure, variables
and instruments in Figure 7 is also the degree to which CLIMCAPS observing capability vary. Overall, CLIMCAPS-Aqua
variation for T and H₂O is significantly higher than that for CLIMCAPS-NOAA20. Given that T is retrieved from CO₂ sensitive
infrared channels, note how CLIMCAPS-NOAA20 AKD for T has insignificant vertical variation across this latitudinal zone,
with an absence of a distinct peak in the troposphere, but its AKD for CO₂ not only has high variability but also a distinct peak





in the upper troposphere. CLIMCAPS-Aqua, on the other hand, has T AKDs with high variability and a distinct tropospheric peak but its $CO_2$ AKDs have no distinct peak and low vertical variability. This suggests that observing capability for $CO_2$ is enhanced (depressed) when it is observing capability for T is depressed (enhanced). Another set of variables that are spectrally correlated is $H_2O$ and $CH_4$. The channels sensitive to $CH_4$ absorption is also sensitive to $H_2O$. CLIMCAPS minimizes their

correlation in the final retrieval products through channel selection for spectral purity coupled with a sequential propagation of scene-dependent uncertainty, but a degree of correlation persists as seen in Figure 7. We see this in CLIMCAPS-NOAA20 observing capability that is lower for both $H_2O$ and $CH_4$ while in CLIMCAPS-Aqua it is higher for both variables.

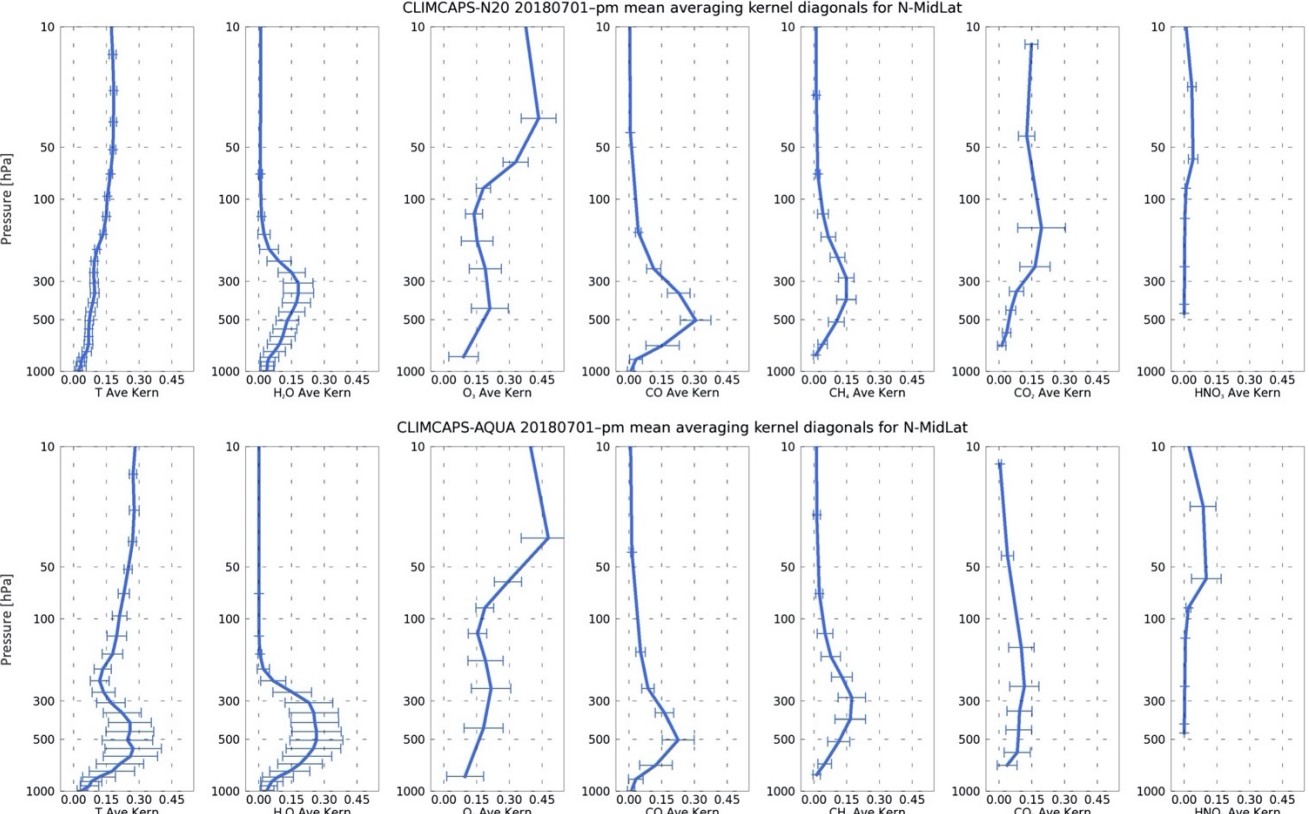

**Figure 7: The mean (blue line) and standard deviation (blue error bars) of averaging kernel matrix diagonals in the Northern Mid-Latitude zone (30˚N to 60˚N) on 1 July 2018 and from (top) CLIMCAPS-NOAA20 and (bottom) CLIMCAPS-Aqua, both ascending orbits. The error bars indicate the degree to which the averaging kernel diagonals vary spatially across the latitudinal zonal.**

### 3.2 Averaging kernels in data inter-comparison studies

Data assimilation models typically use infrared radiance channels to assimilate T and $H_2O$, but for trace gases they use the

retrieved profiles (Levelt et al., 1998; Clerbaux et al., 2001; Yudin, 2004; Segers et al., 2005; Pierce et al., 2009; Liu et al., 2012). Top of atmosphere radiances are highly correlated, highly mixed signals of atmospheric variables. A single channel in



the ~2100 cm$^{-1}$ spectral range may contain information about CO, but it also contains information about N$_2$O, T, surface emissivity, surface temperature and H$_2$O. If a model wants to assimilate CO spectral channels then it would have to account for all interfering species in addition to the uncertainty of CO, least it introduces bias in its characterization of CO processes.

This has proven prohibitively difficult in the case of trace gases where the target variable has a weak spectral signal with interference from variables with much stronger signals. Instead, modellers rely on retrieval algorithms to decompose the infrared channels into distinct trace gas speces. Maddy and Barnet (2008) gave a detailed description of how AKDs can be used together with the retrieved profiles to remove a-priori information from the retrieval and thus facilitate their assimilation at a minimum cost to the model. Today, the Maddy-Barnet method is well established and widely used as the standard method

for data assimilation of retrieved trace gas profiles (Pierce et al., 2009).

In this section, we turn our attention to the value of AKMs in data inter-comparison studies and specifically the inter-comparison of different remote sounding products, all with their own sets of AKMs. What can we learn about a retrieval product from its AKMs and how can this facilitate understanding and interpretation?

Figure 8 illustrates CLIMCAPS-NOAA20 O$_3$ retrieval diagnostics at three different scenes in the Northern Hemisphere

on 1 July 2018. For each scene, the diagnostics are, (i) the O$_3$ averaging kernels and, (ii) the departure from the a-priori (retrieval minus a-priori). The former characterizes CLIMCAPS observing capability for O$_3$ at that scene, and the latter quantifies the changes made to the a-priori, given the measurement information content in the CLIMCAPS channel subset. Recall that CLIMCAPS employs MERRA2 as a-priori for T, H$_2$O and O$_3$ (Smith and Barnet, 2019). MERRA2 assimilates partial column ozone from a series of SBUV instruments between 1980 and September 2004. After September 2004, SBUV

data are replaced by total ozone retrievals from the Ozone Monitoring Instrument (OMI) and stratospheric ozone profiles from MLS (Levelt et al., 1998) on board the NASA Aura satellite. Wargan et al. (2017) validated MERRA2 ozone against ozonesondes and found them to give an accurate representation of cross-tropopause gradients and variability on daily and interannual time scales. MERRA2 does not assimilate any infrared channels or retrievals from CrIS or AIRS for its O$_3$ product. Figure 8 illustrates that CLIMCAPS has observing capability for stratospheric as well as tropospheric ozone, which means it

has the potential to add new information to the MERRA2 a-priori fields in two distinct parts of the atmosphere. While CLIMCAPS-NOAA20 observing capability is similar at all three scenes, we see that the retrieval deviation from the a-priori (black line) varies significantly from scene to scene. In scene a, CLIMCAPS-NOAA20 increased the stratospheric concentrations, while decreasing tropospheric O$_3$. In scene b, CLIMCAPS-NOAA20 mainly reproduced MERRA2 tropospheric O$_3$, while increasing it slightly in the lower stratosphere. In scene c, CLIMCAPS-NOAA20 added no new

information to MERRA2 stratospheric O$_3$, but it increased its upper tropospheric concentrations.

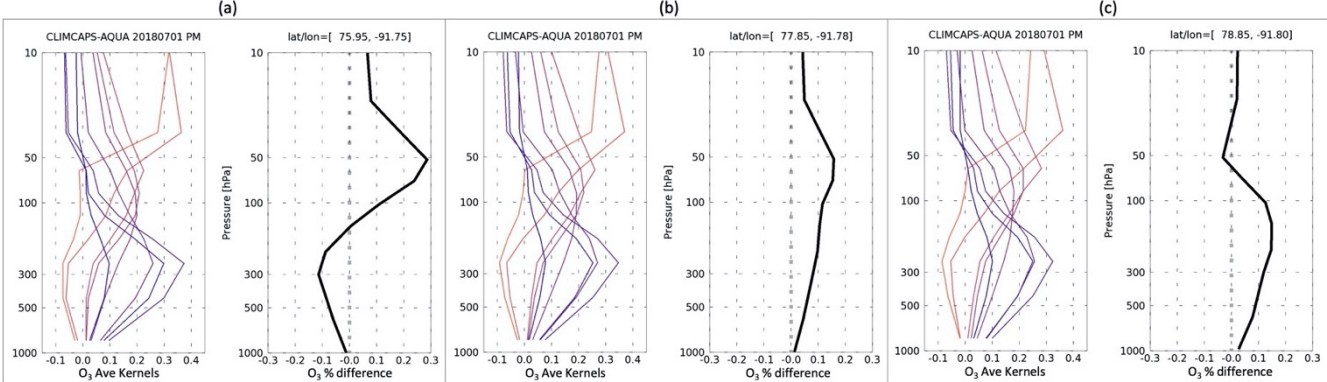

**Figure 8: An evaluation of ozone (O₃) retrievals from CLIMCAPS-NOAA20 ascending orbit on 1 July 2018 for three scenes at (a) 76.0N, 91.8W, (b) 77.9N, 91.8W and (c) 78.9N, 91.8W. For each scene, the averaging kernels are displayed in the left panel and the retrieval departure from a-priori in the right panel. CLIMCAPS uses MERRA2 as a-priori for O₃. Scenes with averaging kernels similar in structure can have an a-priori departure that varies in structure. All three scenes presented here passed CLIMCAPS quality control and are labelled as "successful". For each scene, CLIMCAPS additionally derives uncertainty metrics about the presence of clouds and we list them here. Scene (a) has a cloud fraction (CF) of 1%, cloud top pressure (CTP) of 425 hPa, cloud clearing uncertainty (CCunc) of 0.29 and cloud clearing error (CCerr) of 0.5. Scene (b) has CF=1%, CTP=273 hPa, CCunc=0.29 and CCerr=0.5 and finally scene (c) has CF=3%, CTP=375 hPa, CCunc=0.33 and CCerr=0.76.**

What does it mean when the AKMs show strong observing capability but the retrieval hardly deviates from the a-priori? We interpret this as the CLIMCAPS CrIS IR channel set for O₃ largely confirming the MERRA2 O₃ profile at that scene. Aside from water vapor, ozone is the only trace gas variable in CLIMCAPS that uses an a-priori with space-time structure. All other gases – CO, CO₂, CH₄, N₂O and HNO₃ – use static climatologies. Any space-time structure thus visible in the retrievals of these species originate from the information content in the IR channels only.

For the same day, Figure 9 illustrates CLIMCAPS-NOAA20 temperature retrieval diagnostics for three cloudy scenes in the Southern Hemisphere. Again, we note how the system has similar observing capabilities at each scene, but the retrieval departure from MERRA2 varies significantly. Note how CLIMCAPS-NOAA20 increases MERRA2 temperature at all scenes in the lower stratosphere and troposphere, but decreases MERRA2 temperature in the upper stratosphere. Unlike for O₃, MERRA2 does assimilate CrIS and AIRS IR radiances for its T and H₂O products. This means that CLIMCAPS-NOAA20 adds information from CrIS to MERRA2 for a second time. A few important points are worth noting, though: (i) CrIS and AIRS are two of many sources MERRA2 assimilates for its T and H₂O products, so CrIS/AIRS overall have a weak influence, (ii) MERRA2 does not assimilate CrIS/AIRS channels for cloudy scenes, and (iii) at any given retrieval scene, the CrIS/AIRS assimilated radiances in the MERRA2 products are not from the same orbit of data CLIMCAPS uses in its retrievals.





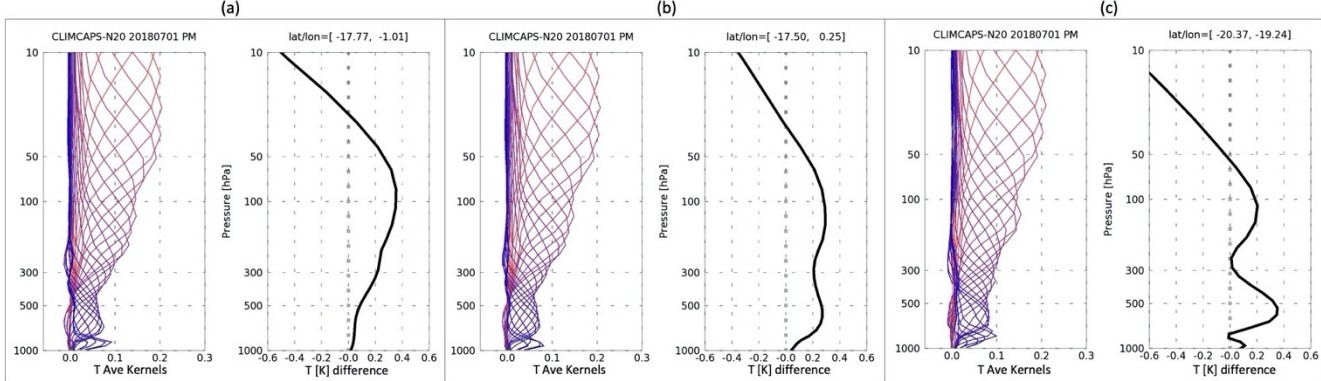

**Figure 9: An evaluation of temperature (T) retrievals from CLIMCAPS-NOAA20 ascending orbit on 1 July 2018 for three scenes at (a) 17.8S, 1.0W, (b) 17.5S, 0.25E and (c) 20.4S, 12.2W. For each scene, the averaging kernels are displayed in the left panel and the retrieval departure from a-priori in the right panel. CLIMCAPS uses MERRA2 as it's a-priori for T. Scenes with averaging kernels similar in structure can have an a-priori departure that varies in structure. Similar to Figure 7, we list the cloud uncertainty metrics for each scene: (i) CF=7%, CTP=175 hPa, $CC_{unc}$=0.18 and $CC_{err}$=1.3 (ii) CF=8%, CTP=158 hPa, $CC_{unc}$=0.15 and $CC_{err}$=1.34 and (iii) CF=0%, $CC_{unc}$=0.12 and $CC_{err}$=0.7.**

When we generate these diagnostic metrics – AKMs and a-priori departure – for CLIMCAPS-NOAA20 retrievals for all scenes from a global day of retrievals, four scenarios emerge, (1) high observing capability with small a-priori departure, (2) high observing capability with large a-priori departure, (3) low observing capability with small a-priori departure, and (4) low observing capability with large a-priori departure. We illustrate this in Figure 10 for CLIMCAPS-NOAA20 retrievals of $H_2O$ on 1 July 2018. For the sake of simplicity, we plot only the AKDs (blue line). The empirically derived thresholds for each metric is 0.1 for AKD and 0.2 for a-priori departure. Scenario 1 (Figure 10, top left quadrant) occurs in ~17% of all CLIMCAPS-NOAA20 retrieval cases, scenario 2 (Figure 10, top right) occurs in 79.5% of all cases, scenario 3 (Figure 10, bottom left) in 1.2% of all cases and scenario 4 (Figure 10, bottom right) in 2.1% of all cases. We calculated these statistics for all retrieval scenes, irrespective of whether the retrievals converged to a solution or not because AKMs are independent of the retrieved variable. CLIMCAPS-20 retrievals flagged as "failed" occur most often in scenarios 3 and 4, where the observing capability is low. These results are summarized in Table 3.

**Table 3: A tabulated summary of the four CLIMCAPS retrieval scenarios.**

| Scenarios | Small a-priori departure | Large a-priori departure |
|---|---|---|
| High observing capability (AKDs) | (1) 17% | (2) 79.5 |
| Low observing capability (AKDs) | (3) 1.2% | (4) 2.1% |



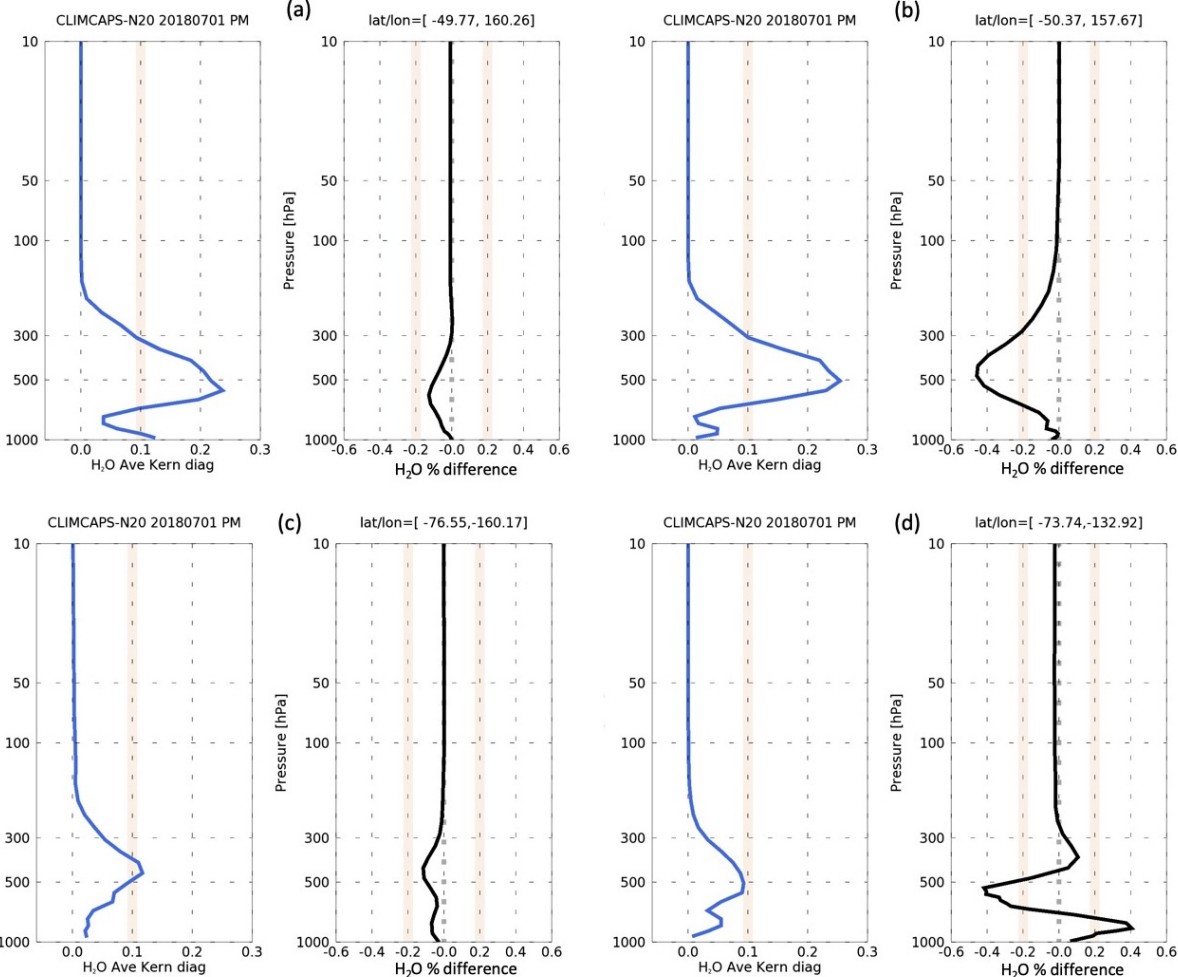

**Figure 10: Towards a generalized diagnostic analysis of CLIMCAPS-NOAA20 retrievals on 1 July 2018. We can broadly identify four different scenarios for CLIMCAPS water vapor (H$_2$O) retrievals by paring the averaging kernel matrix diagonal (AKD; blue line) and retrieval departure (black line) calculated as percent difference: (A-priori minus Retrieval)/(A-priori). AKD is a metric for observing capability. The CLIMCAPS H$_2$O a-priori is MERRA2 so the retrieval departure signifies a disagreement with measured radiances at a target scene. CLIMCAPS scenario (a) has strong observing capability and a small retrieval departure. Scenario (b) has strong observing capability and large retrieval departure. Scenario (c) has low observing capability and small departure. Scenario (d) has low observing capability and large departure. We empirically define the threshold for observing capability as 0.1, and for % difference (a-priori departure) as 20%.**

Data validation studies typically compare remote observations against dedicated aircraft and/or in situ measurements to derive a statistical estimate of overall product accuracy (Nalli et al., 2018a, 2018b). While validation studies are critically important to determine mission objectives, they typically do not provide information on the accuracy of individual soundings from day to day. In science and operational applications, researchers regularly query individual soundings in their study of atmospheric processes and wants to know how well a remote sounding represents the true atmospheric state at a specific scene. Radiosondes are launched daily, but from a sparse network of sites and is thus insufficient in determining site-specific accuracy



for the thousands of satellite soundings each day. In Figure 10, we introduced the four scenarios that emerge when pairing two CLIMCAPS metrics – a-priori departure and magnitude of AKDs – to propose them as a means to help facilitate product interpretation and characterization in the absence of 'truth' data. They can help distinguish those cases where a CLIMCAPS retrieval either departed from or stuck to its a-priori due to higher sensitivity to the true state (large AKDs). A data user can

have confidence that such cases are good representations of the true state. Alternatively, those cases with small a-priori departures and small AKDs (scenario 3) should be interpreted with caution, because the measurements lack the means (information content) with which to confirm or improve upon the a-priori towards a better representation of the true state. Lastly, those retrievals with large a-priori departures and low AKDs (scenario 4) should be rejected as a misrepresentation of the true state because the retrieval is mostly likely dominated by noise, not signal. The a-priori may itself be close to the truth,

but we cannot confirm this due to the system's inability to observe conditions at that scene.

CLIMCAPS has a series of quality control thresholds at various retrieval steps to test T and $H_2O$ retrievals but has no such tests for trace gas variables specifically. As a post-processing step within data applications, the quality control tests are assembled into a data filter that removes unsuccessful T and $H_2O$ retrievals or those with high uncertainty. Currently, the same filters are applied to all retrieved variables, with no distinction made between different variables at a target scene. We propose

here a method with which to diagnosed CLIMCAPS retrievals on a case by case basis, one retrieval variable at a time. Instead of applying a blanket data filter, we illustrate how four diagnostic scenarios (Figure 10, Table 3) can help a data user to characterize retrieval quality along its vertical axis, from boundary layer to top of atmosphere. In Figures 11 and 12 below we build on this to illustrate how these scenarios also apply to CLIMCAPS retrievals along their horizontal axis, i.e., spatially across a swath of observations.

Figures 11 and 12 each have four panels; (a) a-priori departures, calculated as percent difference between CLIMCAPS retrieval and its MERRA2 a-priori at 500 hPa; (b) CLIMCAPS $H_2O$ AKDs at 500 hPa as metric of information content; (c) cloud clearing uncertainty quantified as the 'amplification factor' (Chahine, 1977); and (d) cloud fraction retrievals for each CrIS footprint (or field of view). Figure 11 is a daytime scene (~13h30 local overpass time) over the Caribbean Ocean, including parts of northern Columbia and Venezuela, while Figure 12 is a night-time scene (~01h30 local overpass time) over

the southeast continental United States. Note how CLIMCAPS retrieval departures do not appear spatially random, but instead clustered into distinct features. This means that CLIMCAPS adds new spectral information to its MERRA2 a-priori under specific conditions, which we can diagnose to determine information content and quality. Comparing panels (a) with (c) and (d), we see that there is no direct correlation between retrieval departure (difference between retrieval and a-priori) and presence of or uncertainty due to clouds. This means that CLIMCAPS does have the ability to separate spectral information about $H_2O$ from clouds and add this to its a-priori where necessary. In Figures 11 and 12 we highlight specific features for

discussion – solid lines indicate retrievals that passed all quality control tests and labelled 'good', while dashed lines indicate retrievals that failed at least one quality control test and labelled 'bad'.



**Figure 11: Diagnostic evaluation of CLIMCAPS-NOAA20 retrievals of H2O for ascending Granule 89 (~13h30 local overpass time) on 1 July 2018 over the Caribbean Sea as well as Northern Columbia and Venezuela. (a) H2O retrieval difference as percent departure from a-priori, MERRA2, at 500 hPa. (b) Averaging kernel matrix diagonal vector at ~500 hPa (AKD). (c) Cloud Clearing (CC) amplification factor, a metric of uncertainty about clouds in the radiance signal. (d) Cloud fraction [%] retrieved for each CrIS field of view. Shapes with solid lines indicate scenes where CLIMCAPS retrievals passed all quality control tests and shapes with**
**dashed lines indicate scenes where CLIMCAPS retrievals failed at least one quality control test and are flagged as 'bad'. We label each shape according to the scenario as depicted in Table 3. Shape 2 (scenario 2) has large a-priori departure and large information content. Shape 4 (scenario 4) has large a-priori departure and low information content. Shape 1 (scenario 1) has small a-priori departure and high information content. Panels (c) and (d) provide additional diagnostic information about cloud cover and uncertainty.**


**Figure 12: Same as Figure 11 but for descending Granule 40 (~01h30 local overpass time) on 1 July 2018 over the Southern United States. (a) H$_2$O retrieval difference as percent departure from a-priori, MERRA2, at 500 hPa. (b) Averaging kernel matrix diagonal vector at ~500 hPa (AKD). (c) Cloud Clearing (CC) amplification factor, a metric of uncertainty about clouds in the radiance signal. (d) Cloud fraction [%] retrieved for each CrIS field of view. We highlight features where CLIMCAPS retrievals depart from**
**MERRA2 (a-priori) to demonstrate the diagnostic scenarios introduced in Figure 10. Regions with solid lines indicate scenes where CLIMCAPS retrievals passed all quality control tests and regions with dashed lines indicate scenes where CLIMCAPS retrievals failed at least one quality control test and are flagged as "bad". We label each shape according to the scenario as depicted in Table 3. Shape 4 (scenario 4) has large a-priori departure and low information content. Shape 3 (scenario 3) has small a-priori departure and low information content. Shape 1 (scenario 1) has small a-priori departure and high information content. Shapes 2 (scenario 2)**
**has large a-priori departure and high information content. Panels (c) and (d) provide additional diagnostic information about cloud cover and uncertainty.**





In Figure 10 we use empirically defined thresholds to categorize retrievals into one of four scenarios, 0.1 for AKD and 0.2 for retrieval departure. Figures 11 and 12 demonstrate how they manifest spatially for specific features. Scenario 1, as a small a-priori departure with high information content, is featured in (i) Figure 11 (shape 1) where the region has low cloud clover

(<20% cloud fraction) and very low cloud clearing uncertainty and, (ii) Figure 12 (shape 1) with varying cloud cover that exceed 60% at times, but maintains a relatively low cloud clearing uncertainty. In both these cases, retrievals passed CLIMCAPS quality control, maintained high information content and low cloud uncertainty, so they can be used in applications with confidence and be interpreted as a confirmation of the MERRA2 values for mid-tropospheric moisture. Scenario 2, as a large a-priori departure with high information content, is featured in (i) Figure 11 (shape 2) where CLIMCAPS

retrievals increase MERRA2 $H_2O$ values at 500hPa by as much as 30% and despite significant cloud cover, maintain low cloud uncertainty and, (ii) Figure 12 (shape 2, centred at 35˚N, 97.5˚W) where CLIMCAPS either increases MERRA2 by 10% over a large region and as much as 40% at a localised site where cloud cover and uncertainty are both low; (iii) Figure 12 (shape 2 centred at 29˚N, 98˚W) where CLIMCAPS decreases MERRA2 mid-tropospheric moisture by 20%. In these cases, retrievals passed quality control, maintained high information content in scenes with low cloud cover, so they can be used with

confidence and interpreted as a legitimate departure from MERRA2 and more accurate representation of the true state compared to MERRA2 alone. Scenario 3, as small a-priori departure with low information content, is featured in (i) Figure 12 (shape 3) where information content is below the 0.1 threshold and retrieval departure below 20%. These are retrievals that also failed CLIMCAPS quality tests (indicated by the dashed lines) but for reasons other than cloud uncertainty (which is low) and cloud cover (cloud clearing has high accuracy in partly cloudy scenes such as these). Scenario 4, as large a-priori departure

with low information content, is featured in (i) Figure 11 (shape 4) and (ii) Figure 12 (shape 4) where CLIMCAPS reduces MERRA2 $H_2O$ values at 500 hPa by more than 50% and information content is less than the 0.1 threshold. A very high cloud clearing uncertainty (> 8 amplification of noise) and near solid cloud deck (> 80% cloud fraction) help explain why these retrievals failed quality control tests and should not be trusted in applications. Retrievals with information content less than 0.1 give us no information on the quality of MERRA2 values (we cannot confirm or deny that they correspond with top of

atmosphere measured radiances and therefore know nothing about their accuracy), they only highlight that observing capability was low at that scene. We can diagnose this lack of observing capability, which in itself yields information about the atmospheric state such as cloud cover and uncertainty, but we cannot use the retrievals with any confidence in applications or scientific analyses. On any given global day, a significant majority of the CLIMCAPS retrievals fall into scenarios 1 and 2, which means that we can use them with confidence and interpret their departure from MERRA2 (or lack thereof) with

confidence. Note that the spatial patterns depicted in panels (a) and (b) of Figures 11 and 12 are unique to each retrieval variable and vary with pressure layers according to the AKD shape and vertical profile differences between retrieval and a-priori.



## 4 Summary and Conclusion

In this paper we described our implementation of the Rodger's (2000) Bayesian OE inversion method for CLIMCAPS V2
with a specific focus on averaging kernels. We contrast the Rodgers method for averaging kernels (Eq. 1) with our CLIMCAPS
implementation (Eq. 2) and described the impact our approach has on retrieved products. CLIMCAPS is the NASA system for
generating a continuous record of satellite soundings from two different instrument suites on multiple satellite platforms,
AIRS/AMSU on Aqua and CrIS/ATMS on SNPP and NOAA20. CLIMCAPS products are publicly available through the
NASA EOSDIS Earthdata portal, and each product file contains the full averaging kernel matrix (AKMs) for seven retrieval
variables – T, $H_2O$, CO, $CH_4$, $CO_2$, $O_3$ and $HNO_3$ at every scene. CLIMCAPS AKMs vary in shape and magnitude across
(i) retrieval variables according to top-of-atmosphere spectral sensitivity and instrument spectral resolution, (ii) satellite
platforms according to instrument characteristics and retrieval algorithm assumptions and, (iii) retrieval scenes according
instrument effects such as view angle, environmental conditions such as temperature lapse rates, uncertainty in interfering and
background variables, as well as a-priori assumptions about the target variable. At any given scene, the AKM for one variable
is largely independent from that of another due to the CLIMCAPS sequential retrieval approach (Table 1; Smith and Barnet
2019) and infrared channel selection to minimize spectral interference. For the first time, we compare the observing capability
from CLIMCAPS-Aqua with CLIMCAPS-NOAA20 to diagnose and characterize continuity in information content across
satellite platforms and instrument technology. In summary, we can state the following:

-   The observing capability for T and $H_2O$ is different between CLIMCAPS-Aqua and CLIMCAPS-NOAA20. This
may be due to differences in how we regularize the OE solution for each satellite suite of instruments, but may also reflect
fundamental instrument differences – AIRS on Aqua is a grating spectrometer and CrIS on NOAA20 a Michelson
interferometer. In future, we will investigate this question.

-   CLIMCAPS-NOAA20 has a higher observing capability for $CO_2$ in the mid-troposphere than CLIMCAPS-Aqua.

-   CLIMCAPS has peak observing capability for CO and CH4 in the mid-troposphere, with CO at ~500hPa and CH4
at ~300-400hPa.

-   CLIMCAPS information content for T, $H_2O$, CO and $O_3$ are largely independent of each other with different spatial
patterns in their derived DOF (trace of AKM).

-   CLIMCAPS-NOAA20 has latitudinal variation in its observing capability for $H_2O$, $O_3$, CO, $CH_4$, $CO_2$. For $H_2O$,
CLIMCAPS-NOAA20 observing capability peaks in the tropics (30˚S to 30˚N) at 300 hPa, while it peaks lower down at
450 hPa outside of the Tropics. CLIMCAPS-NOAA20 has the highest latitudinal variability for $O_3$ observing capability with
strongest peaks in the Tropics, both in the stratosphere and troposphere. CLIMCAS-NOAA20 has almost no vertical
stratification in observing capability in the polar regions (> 60˚N and < 60˚S). The mid-latitude regions have $O_3$ AKM peaks
in the stratosphere only. $CO_2$ AKMs have strongest peak at 200 hPa in the tropics. Tropical $CH_4$ has much lower vertical
resolution (as seen in its broad averaging kernel functions) with no distinct peak at 400 hPa as seen in other latitudinal zones.



-    CLIMCAPS-Aqua has latitudinal variation in its observing capability for T, $H_2O$, $O_3$, $CH_4$, and $HNO_3$. It is lowest in the boundary layer for all variables. It has highest vertical resolution (sharpest peak) for T at 700 hPa in the North Polar region (> 60˚N). CLIMCAPS-Aqua has lower observability for tropospheric $O_3$ in the Tropics. $HNO_3$ AKMs have distinct latitudinal variation with highest observability in the stratosphere (< 100 hPa) for all zones, but strongest in the North Polar regions (> 60˚N) followed by Mid-Latitudes, South Polar and Tropics in that order.

-    CLIMCAPS, whether from NOAA20 or Aqua, has sensitivity to $O_3$ and $CO_2$ in two broad layers, one in the mid-troposphere and another in the stratosphere (< 50 hPa), sensitivity to CO and $CH_4$ in one broad mid-tropospheric layer, $HNO_3$ in one broad stratospheric layer, and multiple narrow tropospheric layers for $H_2O$ and T, with additional layers in the stratosphere for T.

We identified four scenarios with which to diagnose CLIMCAPS retrievals vertically along a pressure gradient on a scene-
by-scene basis. These scenarios are, (1) high observing capability (large AKD) and small a-priori departure, (2) high observing capability (large AKD) with large a-priori departure, (3) low observing capability (small AKD) with small a-priori departure, and (4) low observing capability (small AKD) with large a-priori departure. CLIMCAPS has additional uncertainty metrics for evaluating retrievals, such as cloud clearing amplification factor, radiance residual, cloud fraction and cloud top height, DOF, retrieval covariance error, convergence strength and whether a range of quality control thresholds were exceeded. As a
long-term record of temperature, moisture and trace gases, that is continuous and consistent across instruments and satellite platforms, CLIMCAPS V2 products can be useful in characterizing diurnal and seasonal atmospheric processes from different time periods and regions across the globe.

**Data Availability**

CLIMCAPS data products are publicly available through the NASA Earth Observing System Data and Information System
online portal ([https://earthdata.nasa.gov/eosdis](https://earthdata.nasa.gov/eosdis)).

**Author Contribution**

Conceptualization, both authors.; methodology, C.D.B.; software, both authors; formal analysis, N.S.; investigation, both authors; writing—original draft preparation, N.S.; writing—review and editing, N.S.; visualization, both authors; funding acquisition, C.D.B. (Principal Investigator) and N.S. (Co-Investigator).

**Acknowledgements**

This research was funded by NASA, grant award number 80NSSC18K0975, and our work described here benefits from rich discussions with a wide range of colleagues. We wish to thank them all.



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
