# Peer review of "CLIMCAPS Observing Capability for Temperature, Moisture and Trace Gases from AIRS/AMSU and CrIS/ATMS"

_Atmospheric Measurement Techniques, 2020_

## Referee Comment (RC1) · Anonymous Referee #1 · 10 Apr 2020

This paper is an excellent contribution to the application of hyperspectral infrared remote sounding data. The CLIMCAPS algorithm was selected by competitive proposal and data are now reported to be available to the public. This paper serves to introduce users to some novel features of the dataset. The authors discuss several ways the data may be analysed particularly with regard to information content. The examples provided only scratch the surface of the possible applications but are a useful guidepost for users. The comparison of the technical details of the CLIMCAPS algorithm with the widely known Rodgers approach is very helpful. The discussion of averaging kernels is thorough and figures 11 and 12 provide real life examples of the concepts. The paper is appropriate for AMT and I recommend publication with minor revisions as

described in the following paragraphs, first as questions for the authors to consider and then some technical corrections.

Questions for the authors (all line numbers refer to the manuscript as originally posted):

1. Line 25. The statement from Smith, 2013, appears controversial if it implies we now have adequate data for earth systems analysis. Those of us who have proposed instruments since 2013 might dispute this. I suggest the authors either (a) add some context from the original paper (did they mean we have sufficient but inaccurate data, or data sufficient for some purposes but not others?) or (b) comment on this statement from the perspective of 2020 and CLIMCAPS.

2. Figure 1b. Is the CrIS-NPP noise for NSR or FSR, or does it matter?

3. Line 167 and 507. Are any data assimilated by MERRA2 always from a previous orbit? Then data are never absent because of cloudy scenes in this orbit if they would not be considered anyway and the cloudy criterion seems superfluous.

4. Line 171 and 498. Are these climatologies single valued profiles for all space and time, or do they have latitudinal dependence? Is a single $CO_2$ profile used for all time, and does this make retrievals at one time favoured over another? It would be helpful to have a reference to the climatologies used.

5 Fig 2 caption. Likewise it would be helpful to have a reference to the Masuda model as there may be multiple versions thereof.

6. Line 199 and Table 1. The text implies that $N_2O$ and $SO_2$ will be in the table and they are not.

7. Line 410. This seems the key point of the continuity mission. The authors have been admirably frank (line 441) in discussing some minor shortcomings of the present version. How would this re-evaluation be done? What will you look at? In particular are there theoretical criteria which can be used? What would be a success criterion for a continuity product?

8. Figure 6(d). The low DOF for ozone over Canada is an interesting feature not discussed in the text. Is it due to low ozone values, low temperatures, stratospheric warming, or something else? A sentence about the physical state would demonstrate the utility of the DOF analysis.

Technical corrections:

Line 10. Remove "individually" or rewrite the sentence as it is confusing

Line 159. hamper -> hampers

Line 270. least their uncertainty is -> lest their uncertainty be

Line 443. Remove "it is"

Line 444. absorption is -> absorption are

Line 459. least it introduces -> lest it introduce

Fig 9 caption. it's -> its

Fig 10 caption. paring -> pairing

Line 544. wants -> want

Line 545. and is thus -> and are thus

Line 566 and Fig 11 caption. Columbia -> Colombia

Line 605. exceed -> exceeds

Line 611. centred -> centered

Line 612. localised -> localized (the rest of the document is US spelling)

Line 634. No apostrophe in Rodgers

Line 661. CLIMCAS -> CLIMCAPS

[Figure]

Line 714. Retemote -> Remote

Line 829. propagation systematic -> propagation of systematic

---

## Referee Comment (RC2) · Anonymous Referee #2 · 2 May 2020

This is the first review of the manuscript titled "CLIMCAPS Observing Capability for Temperature, Moisture and Trace Gases from AIRS/AMSU and CrIS/ATMS" by N. Smith and Ch. Barnet. The paper is very well written and is easy to follow. The Figures are well designed and support the reading of the paper. It has been structured to methodically investigate the optimization of the common retrieval algorithm applied to different sensors. Several important climatological products are produced and their independence is assessed from the spectroscopic and scene dependent approach. The a priori information in the retrieval is replaced by the damping factor that is a variable and dependent on the retrieved parameter and the scene. The example of the relation between the eigenvalues and damping factors is very useful for the reader.

[Figure]

The damping coefficient is also investigated to compare different scenes and changes in the cloud clearing algorithm. The algorithm is set to get the most information possible from each retrieval but also avoids over constraining the solution to the a priori where no additional information can be added from the observation. Many examples are provided throughout the text to demonstrate sensitivity in the retrieval of various climatological and dynamically changing parameters (i.e cloud clearing). Authors developed the quality control approach where each retrieval is assessed based on the departure from a priori, information content of the AKD, and uncertainties associated with the cloud clearing algorithm. Still, high variability in the retrieval approach can also lead to long-term and spatial changes in the vertical sampling of the atmosphere. Here are the general questions. 1) Since the damping parameter change for each scene, the AKM also changes. Can it impact the vertical smoothing and therefore alter the effective altitude of the nominal layer in the retrieved profile? Assuming that over time the scene over a particular geographical location might change (i.e. due to climate impact) and it could lead to changes in the altitude contribution to a particular layer in the retrieved profile. So, effectively the long term trends could be impacted by the change in the altitude where most of the information comes from? 2) If AK shapes are very different between instruments located at AIRS and JPSS, how are you proposing to combine records in the long-term time series? 3) If uncertainties change over each scene, are these saved for each retrieval and provided for creating the gridded products?

Specific questions. 1) Lines 248-260 – does reduction in the vertical resolution of the retrieved profiles lead to the issues with the interpolation of profiles for the iterations in the forward model that has 100 layers? Was this error investigated? 2) Lines 454-508, Section 3.2 discusses the interpretation of the informational content of profiles from adjacent scenes. The MERRA-2 is used as a priori and sometimes the RT returns the a priori. Ozone, H2O, and temperature RTs are the only ones that use MERRA2 a priori that changes with time and space (other species are retrieved with static a priori climatology). How much could it impact on the derived ozone and H2O trends?

[Figure]

AK could be the same, but a priori could change with time.... Have you assessed the trends in MERRA2 ozone a priori, particularly in the troposphere where ozone variability is limited to the differential column between assimilated stratospheric ozone from MLS and OMI total column? Is $H_2O$ a priori changing in the upper troposphere since 2002 and how it might be related to the tropopause variability? 3) Lines 517-528 – Discussion and Table 3 present a summary of the quality of retrieval results for one day. The Figure 10 shows four examples at different latitudes. A number of profiles that have high observing capability are significantly higher than for low sensitivity cases. For the profile to differ from a priori significantly or not depends on the a priori (as you mentioned). In the case of $H_2O$ the a priori is climatology and thus the retrieval wit high observing capability should have a larger departure from a priori. Can you please provide information if this result is common for any other day, and how it changes by scene, location, or time? Also, it would be good to see a priori profiles and AKD for temperature retrievals matching the $H_2O$ examples shown in Figure 10. 4) Lines 575-577 – Figures 11 and 12 show daytime and nighttime retrievals, but not over the same geographical area. Why not? IT would be of interest for the reader to learn about differences between daytime and night time observational sensitivities. Many air quality studies rely on the contrast in ozone and WV levels between nighttime and daytime.

---

## Author Comment (AC1) · 11 Jun 2020

We thank the Reviewer for their positive feedback and the interesting questions they raised. Our responses are below.

General questions:

1) Since the damping parameter change for each scene, the AKM also changes. Can it impact the vertical smoothing and therefore alter the effective altitude of the nominal layer in the retrieved profile? Assuming that over time the scene over a particular geographical location might change (i.e. due to climate impact) and it could lead to

changes in the altitude contribution to a particular layer in the retrieved profile. So, effectively the long term trends could be impacted by the change in the altitude where most of the information comes from?

This is a good summary of the challenge we face when constructing long-term records and why we caution against using satellite sounding records in trend analyses without careful consideration (Smith and Barnet, 2019). Retrieved soundings are underdetermined inverse measurements with dependence on prior knowledge about the atmospheric state. This is true not only for CLIMCAPS, but soundings in general. This dependence on the a-priori varies from scene to scene as the information content in the top of atmosphere radiance measurements vary. We can only retrieve as much information as is available in the instrument measurement of a given scene at a specific time.

The instruments we discuss in this paper were not designed for climate trend studies. They were designed for weather monitoring in (near) real-time. This means that the traditional approach to designing retrieval systems is instrument specific and focused on achieving instantaneous tropospheric accuracy. We now have almost two decades of hyperspectral infrared measurements; AIRS on the Aqua (launched 2002) is near the end of its impressive lifetime and we have the CrIS instrument first on SNPP followed by the JPSS series of satellites (one of which was launched in 2017) with planned availability into the 2040s. With CLIMCAPS, we are building a system to help us, as community, address whether these sounding measurements can add value as a long-term record and help us improve our knowledge of retrieval system design for climate science. Every diagnostic metric we used in this paper, together with the various error sources and quality control filters are available in CLIMCAPS product files distributed by NASA GES DISC (earthdata.nasa.gov). With this product now publicly available we wish to explore the questions the Reviewer raised here. We aim to characterize information content for different geographic regions and atmospheric processes so that we can investigate the quality of information available under different conditions. With a

baseline now established (as communicated in this paper), we can advance our understanding of the value of these satellite measurements outside of weather forecasting.

2) If AK shapes are very different between instruments located at AIRS and JPSS, how are you proposing to combine records in the long-term time series?

With this paper we are presenting our first information content evaluation of the CLIMCAPS-Aqua (AIRS) and CLIMCAPS-NOAA20 (JPSS) systems. We did not expect their averaging kernels (AK) to show such marked differences. Can we optimize AKs to show similar shapes under similar conditions from different instruments? Are these differences due to our algorithm design or fundamental instrument differences? We will research these questions and adjust the algorithm where necessary/possible in CLIMCAPS Version 3 (due for launch in Summer 2021) so that we can achieve consistency and continuity in information content across decades to promote the usefulness of these products in climate studies. At the very least, such consistency should simplify the interpretation of observed changes over time.

This said, we designed the retrieved soundings from CLIMCAPS Version 2 (as presented in this paper) from Aqua (AIRS/AMSU) and JPSS (CrIS/ATMS) to be continuous as a time-series, irrespective of the shapes of their AK by ingesting MERRA2 as a-priori for temperature and water vapor. In optimal estimation inversion, the a-priori simply fills the null space (Identity matrix minus AK) and thus compensates for lack of information content in the measurements. MERRA2 is a well characterized, accurate representation of atmospheric change over two decades.

3) If uncertainties change over each scene, are these saved for each retrieval and provided for creating the gridded products?

Yes, these are all saved with the retrieved soundings in the product file. NASA GES DISC hosts a range of CLIMCAPS product and application guides that explain the available retrieval and uncertainty fields.

The Reviewer raised a critical issue here and that is the degree of consideration one should give to retrieval (Level 2) uncertainty when gridding them (Level 3). Traditionally, Level 3 methods simply average all retrievals that passed pre-defined quality control filters without consideration for scene-specific uncertainty. This is an area of research we think should get more attention so that we can improve the scientific value of gridded (Level 3) products and promote understanding when comparing against other gridded products (e.g, Smith et al. 2013).

Specific questions:

1) Lines 248-260 – does reduction in the vertical resolution of the retrieved profiles lead to the issues with the interpolation of profiles for the iterations in the forward model that has 100 layers? Was this error investigated?

No, because all forward model calculations in CLIMCAPS are performed on 100 layers. The CIMCAPS Jacobians are calculated with brute force perturbation of the a-priori profile, and these are done not as single-layer perturbations but multiple layers as defined by the trapezoid hinge points. The result is a Jacobian (K) matrix with reduced dimension along the state axis. This greatly speeds up the OE retrieval, which is an important consideration for global and decadal data (re)processing. It does introduce an additional factor of smoothing, which was thoroughly documented by (Maddy et al., 2009; Maddy and Barnet, 2008; Susskind et al., 2003). CLIMCAPS builds on the methods developed by the AIRS Science Team from the past three decades, so we did not deem it necessary to investigate the effect of this smoothing on the V2 retrievals. We could revisit it in future if it becomes relevant in a specific application.

2) Lines 454-508, Section 3.2 discusses the interpretation of the informational content of profiles from adjacent scenes. The MERRA-2 is used as a priori and sometimes the RT returns the a priori. Ozone, $H_2O$, and temperature RTs are the only ones that use MERRA2 a priori that changes with time and space (other species are retrieved with static a priori climatology). How much could it impact on the derived ozone and $H_2O$

trends?

MERRA2 will strongly impact trends in CLIMCAPS temperature, H2O and O3 retrievals since MERRA2 fills in the retrieval null space by design.

3) AK could be the same, but a priori could change with time...

Exactly. But this is true for all sounding systems, not just CLIMCAPS.

4) Have you assessed the trends in MERRA2 ozone a priori, particularly in the troposphere where ozone variability is limited to the differential column between assimilated stratospheric ozone from MLS and OMI total column? Is H2O a priori changing in the upper troposphere since 2002 and how it might be related to the tropopause variability?

We have periodic conversations with the MERRA2 team and are aware of the jumps in the record when MLS was introduced and we have had discussions for the plans when MLS data is no longer available. We argue that it is better to have a well characterized a-priori, even if it changes with time, than to have an a-priori that is not well characterized.

The first full record of CLIMCAPS-SNPP and -NOAA20 is now publicly available. The product files contain the CLIMCAPS retrieval, space-time-pressure interpolated MERRA2 a-priori for temperature, H2O and O3, together with the averaging kernels, uncertainty, error and quality control estimates at each retrieval scene. We, as community, can now address the questions the Reviewer posed.

5) Lines 517-528 – Discussion and Table 3 present a summary of the quality of retrieval results for one day. The Figure 10 shows four examples at different latitudes. A number of profiles that have high observing capability are significantly higher than for low sensitivity cases. For the profile to differ from a priori significantly or not depends on the a priori (as you mentioned). In the case of H2O the a priori is climatology and thus the retrieval wit high observing capability should have a larger departure from a priori. Can you please provide information if this result is common for any other day, and how

it changes by scene, location, or time? Also, it would be good to see a priori profiles and AKD for temperature retrievals matching the H2O examples shown in Figure 10.

We are encouraged that our discussion in this paper generates these ideas for experiments worth investigating in future. We are cautious, however, to add more figures and expand the discussion in the paper too much beyond what we have for the sake of efficiency. We are eager to explore and demonstrate all the issues the Reviewer lists here in future work.

Just a side note: the CLIMCAPS a-priori for H2O is MERRA2, not a climatology.

6) Lines 575-577 – Figures 11 and 12 show daytime and nighttime retrievals, but not over the same geographical area. Why not? IT would be of interest for the reader to learn about differences between daytime and night time observational sensitivities. Many air quality studies rely on the contrast in ozone and WV levels between nighttime and daytime.

Our goal in this paper was to introduce CLIMCAPS V2 (the first public release) and additionally illustrate how its retrievals can be diagnosed using the available information content and uncertainty metrics. It will be very interesting to evaluate diurnal effects using the technique presented in Figures 11 and 12, and we will explore this in future work.

References

Maddy, E. S. and Barnet, C. D.: Vertical Resolution Estimates in Version 5 of AIRS Operational Retrievals, IEEE Transactions on Geoscience and Remote Sensing, 46(8), 2375–2384, doi:10.1109/TGRS.2008.917498, 2008.

Maddy, E. S., Barnet, C. D. and Gambacorta, A.: A Computationally Efficient Retrieval Algorithm for Hyperspectral Sounders Incorporating A Priori Information, IEEE Geoscience and Remote Sensing Letters, 6(4), 802–806, doi:10.1109/LGRS.2009.2025780, 2009.

Smith, N. and Barnet, C. D.: Uncertainty Characterization and Propagation in the Community Long-Term Infrared Microwave Combined Atmospheric Product System (CLIM-CAPS), Remote Sensing, 11(10), 1227, doi:10.3390/rs11101227, 2019.

Smith, N., Menzel, W. P., Weisz, E., Heidinger, A. K. and Baum, B. A.: A Uniform Space–Time Gridding Algorithm for Comparison of Satellite Data Products: Characterization and Sensitivity Study, Journal of Applied Meteorology and Climatology, 52(1), 255–268, doi:10.1175/JAMC-D-12-031.1, 2013.

Susskind, J., Barnet, C. D. and Blaisdell, J. M.: Retrieval of atmospheric and surface parameters from AIRS/AMSU/HSB data in the presence of clouds, IEEE TGRS, 41, 390–409, 2003.
* * *

---

## Author Comment (AC2) · 11 Jun 2020

We would like to thank the Reviewer for their generous review of our paper and their comments encouraging us to revisit, clarify and improve a few important sections.

Questions for the authors (all line numbers refer to the manuscript as originally posted):

1. Line 25. The statement from Smith, 2013, appears controversial if it implies we now have adequate data for earth systems analysis. Those of us who have proposed instruments since 2013 might dispute this. I suggest the authors either (a) add some context from the original paper (did they mean we have sufficient but inaccurate data,

or data sufficient for some purposes but not others?) or (b) comment on this statement from the perspective of 2020 and CLIMCAPS.

On re-reading Line 25, we see how our statement misleads. We meant to argue that the existence of high quality satellite measurements across decades from many instruments on multiple platforms does not in itself imply information consistency or the ability to support climate research. In our paper we argue for data processing methods that pay close attention to uncertainty characterization and rigorously account for variability in measurement information content over time and space. The Smith et al. (2013) paper we cite highlighted the challenges in oversimplifying the issue – merely applying the same retrieval method to hyperspectral infrared measurements from different instruments/platforms does not guarantee geophysical consistency. However, this does not mean that the nearly two decades of satellite sounder measurements lack value in climate studies. We argue here that we have a lot to learn yet about constructing long-term satellite sounder records and characterizing the information they can contribute to Earth system research.

The sentence now reads: "While the record of hyperspectral infrared measurements span nearly two decades, changes in technology and instrumentation pose a significant challenge to data continuity (Smith et al., 2013)."

2. Figure 1b. Is the CrIS-NPP noise for NSR or FSR, or does it matter?

The CrIS-NPP instrument noise we depict in Figure 1b is for NSR, while CrIS-NOAA-20 is for FSR. The difference between these two lines, illustrates the difference in instrument noise between the CrIS NSR and FSR. We added a clarification to the Figure caption and thank the Reviewer for highlighting this shortcoming. The difference between CrIS FSR and NSR is part of our argument here.

3. Line 167 and 507. Are any data assimilated by MERRA2 always from a previous orbit? Then data are never absent because of cloudy scenes in this orbit if they would not be considered anyway and the cloudy criterion seems superfluous.

The simple answer to the Reviewer's questions here is, No. Our statement was misleading and more descriptive of forecast models, such as the Global Forecast System (GFS), than of reanalysis models, such as MERRA2. We updated Section 2.1 (previously line 167) with the following text:

"MERRA2 assimilates a small subset of IR channels (i.e., by selecting channels that are primarily sensitive to T but largely insensitive to H2O, clouds and trace gases) only sometimes (i.e., for clear-sky scenes only) and weigh it based on the time of measurement within the reanalysis window and with an assumed representation error across all scenes. This gives us confidence to argue that the IR channels used in CLIM-CAPS rarely duplicates the information content of the IR channels used in MERRA2 at a specific scene. Stated differently, the IR information content from AIRS or CrIS in CLIMCAPS is much higher than in MERRA2 because CLIMCAPS retrieves the atmospheric state along line of sight, from a greater selection of cloud cleared IR channels (i.e., all scenes except those with uniform cloud cover) and a full accounting of trace gas absorption. We contrast the CLIMCAPS a-priori approach with those systems that employ a regression first guess such as AIRS V6 (Susskind et al., 2014) that runs a non-linear regression using all IR channels to derive it's a-priori for T, H2O and O3. Unlike AIRS V6, CLIMCAPS does not use the full information content of the available IR channels twice to avoid an aliasing of its retrieval null space error and amplification of instrument uncertainty."

And in Section 3.1 (previously Line 507), we revised the discussion about MERRA2 as follows:

"MERRA2 does assimilate CrIS and AIRS IR radiance channels that are sensitive to temperature. We argue, however, that on a scene-by-scene basis it is highly improbable that CLIMCAPS uses IR measurements twice (first as assimilated information in MERRA2, second as measurement vector in OE retrievals) due to the strong spectral and spatial filters adopted in data assimilation systems. Even where a MERRA2 grid cell does contain IR information at a target CLIMCAPS footprint, we consider the impact of the assimilated IR channels on the OE retrieval to be negligible. CLIMCAPS aggregates an array of 3 x 3 fields of view (∼14 km) during cloud clearing (step 3 in Figure 2) and retrieves all subsequent variables from the cloud cleared radiance that represents the clear portion of partly cloudy atmospheres on a larger field of regard (∼50 km). MERRA2, on the other hand, assimilates single field of view radiances for clear-sky atmospheres. MERRA2 assimilates measurements from many sources, so the contribution made by a single source at a target site is low, especially considering that each source is weighed according to a static, pre-determined representation error. CLIMCAPS, on the other hand, uses cloud cleared IR radiances as one of its primary sources of information that it weighs based on scene-specific information content analysis."

4. Line 171 and 498. Are these climatologies single valued profiles for all space and time, or do they have latitudinal dependence? Is a single CO2 profile used for all time, and does this make retrievals at one time favoured over another? It would be helpful to have a reference to the climatologies used.

We addressed this shortcoming by adding the following text to Section 2.1 (previously Line 171).

"For the trace gas species, we adopted the same approach in CLIMCAPS as that used in AIRS V6 for CO, CO2, HNO3, N2O and SO2 (AIRS Science Team/Joao Texeira, 2013). The CO climatology has no intra-annual variation but does vary seasonally and latitudinally, while the COÂň2 climatology is a static value across all latitudes but increases annually according to the linear fit developed by (Maddy, 2007). The climatologies for the remaining trace gas species, HNO3, N2O and SO2, are static over time and space. The CLIMCAPS climatology for CH4 is derived from a set of coefficients developed by (Xiong et al., 2008, 2013) that is also used in the NOAA-Unique Combined Atmospheric Processing System (NUCAPS)."

In Section 3.1 (previously Line 498) we added this: "All other gases – CO, CO2, CH4,

N2O and HNO3 – use climatologies as discussed in Section 2.1"

5. Fig 2 caption. Likewise it would be helpful to have a reference to the Masuda model as there may be multiple versions thereof.

In CLIMCAPS we adopted the ocean emissivity model as implemented in AIRS V6 (AIRS Science Team/Joao Texeira, 2013) which the Masuda, et al., (1988) model as modified by Wu and Smith (1997). We added these references to the caption.

6. Line 199 and Table 1. The text implies that N2ÂňO and SO2 will be in the table and they are not.

Thanks for pointing this out. We added N2O and SO2 to the table for the sake of completion.

7. Line 410. This seems the key point of the continuity mission. The authors have been admirably frank (line 441) in discussing some minor shortcomings of the present version. How would this re-evaluation be done? What will you look at? In particular are there theoretical criteria which can be used? What would be a success criterion for a continuity product?

These are great questions that raise interesting and important issues. There is no simple answer and we think it would be insufficient to rely on the methods traditionally used in data validation with point-source comparisons. We are concerned here with evaluating continuity at much larger scales. The conundrum, of course, is that there is no 'truth' dataset at such scales and the only option being comparison with other datasets that have comparable limitations. This paper by (Gaudel et al., 2018) is a classic example of how data comparison can lead to a 'now what?' moment. What if no two sources agree?

The other challenging aspect is that satellite soundings are inverse measurements with dependence on prior knowledge about the atmospheric state. What if a sounding system uses a model specifically designed for continuity across the AIRS/CrIS era as

a-priori, strongly damps the measurement contribution during the OE retrieval step, thus reproducing the model, and call it a continuity record? Without access to the system's averaging kernels, the retrieval record will be misinterpreted and may even be compared against the very model it used as a-priori. With CLIMCAPS, we make a concerted effort to be transparent about the nature of its inverse measurements (retrievals) to encourage meaningful data comparisons especially across different instruments/platforms.

What would we consider a success criterion for a continuity product? Consistency in information content, and specifically the averaging kernels. The differences we observe in averaging kernels between CLIMCAPS-Aqua and CLIMCAPS-NOAA20 (Figure 5) give us pause. It tells us the two systems apply different weighting to the radiance measurements and thus vary in their dependence on the a-priori. With our next release, we would like to see consistency in vertical sensitivity between CLIMCAPS soundings from the different satellite platforms, not on a scene-by-scene basis, but systematically across latitude zones and seasons. This would tell us that we achieved consistency in observing capability across satellites, with continuity in their sensitivity to the true atmospheric state under similar types of conditions. So, in answering this question, averaging kernels would be our metric and consistency in their shape and magnitude under similar conditions across instruments would be our success criterion. Testing for continuity in retrieval accuracy would require different methods that, for now, falls outside the scope of our efforts.

We expanded on this issue in Section 3.1 (previously Line 410):

"In future, we will experiment with these threshold values to test if we can achieve consistency in averaging kernels across CLIMCAPS-Aqua, -NOAA20 and -SNPP. We are interested in addressing the question whether we can achieve continuity in information content despite instrument differences. The disparity in information content we currently observe between CLIMCAPS-Aqua and CLIMCAPS-NOAA20 (Figure 5) tell us that the two systems apply different weighting to the radiance measurements and thus

vary in their dependence on the a-priori. This can introduce inconsistencies in the data record and hamper continuity. In using averaging kernels as metric, we can evaluate information content under similar conditions across CLIMCAPS-Aqua, -NOAA20 and -SNPP and thus test for continuity in their observing capability."

8. Figure 6(d). The low DOF for ozone over Canada is an interesting feature not discussed in the text. Is it due to low ozone values, low temperatures, stratospheric warming, or something else? A sentence about the physical state would demonstrate the utility of the DOF analysis.

This is an interesting feature, especially since none of the other variables have it. O3 DOF could be low over Canada for any of the reasons the Reviewer listed. Analyzing DOF features from specific variables is beyond the scope of this paper, but we added the following sentence to the discussion of Figure 6 to suggest that it is possible to analyze the physical state alongside the DOF to better understand observing capability:

"Where DOF patterns do have distinct features, such as the low O3 DOF feature over Canada (Figure 6d), we can understand them by evaluating the physical state to determine if it is due to conditions such as low O3 concentrations, low lapse rates or stratospheric warming. All retrieval variables and their uncertainty metrics are coincident in space and time in the CLIMCAPS product files."

9. Technical corrections:

We thank the reviewer for their careful read of our paper. We corrected all the mistakes they listed.

References:

AIRS Science Team/Joao Texeira: Aqua AIRS L2 standard retrieval product using AIRS IR and AMSU, without-HSB V6, doi:10.5067/AQUA/AIRS/DATA201, 2013.

Gaudel, A., et al.: Tropospheric Ozone Assessment Report: Present-day distribution and trends of tropospheric ozone relevant to climate and global atmospheric chemistry

model evaluation, Elem Sci Anth, 6, doi:10.1525/elementa.291, 2018.

Maddy, E. S.: Investigations of the spatial and temporal resolutions of retrievals of atmospheric CO2 from the Atmospheric InfraRed Sounder (AIRS), Ph.D, University of Maryland., 2007.

Smith, N., Menzel, W. P., Weisz, E., Heidinger, A. K. and Baum, B. A.: A Uniform Space–Time Gridding Algorithm for Comparison of Satellite Data Products: Characterization and Sensitivity Study, Journal of Applied Meteorology and Climatology, 52(1), 255–268, doi:10.1175/JAMC-D-12-031.1, 2013.

Susskind, J., Blaisdell, J. M. and Iredell, L.: Improved methodology for surface and atmospheric soundings, error estimates, and quality control procedures: the atmospheric infrared sounder science team version-6 retrieval algorithm, Journal of Applied Remote Sensing, 8(1), 084994, doi:10.1117/1.JRS.8.084994, 2014.

Xiong, X., Barnet, C., Maddy, E., Sweeney, C., Liu, X., Zhou, L. and Goldberg, M.: Characterization and validation of methane products from the Atmospheric Infrared Sounder (AIRS), Journal of Geophysical Research, 113(null), doi:10.1029/2007JG000500, 2008.

Xiong, X., Barnet, C., Maddy, E. S., Gambacorta, A., King, T. S. and Wofsy, S. C.: Mid-upper tropospheric methane retrieval from IASI and its validation, Atmospheric Measurement Techniques, 6(9), 2255–2265, doi:10.5194/amt-6-2255-2013, 2013.

---

## Author Response (AR1)

**Changes Made in Response to Reviewer Comments**

- We made change to the manuscript as outlined in our response to Reviewer 1
- Reviewer 2 did not request changes to the manuscript but instead challenged us with conceptual questions that we addressed as thoroughly as possible.
- We updated the Flow Diagram (Figure 2) and its caption to clarify the retrieval flow.

[revised manuscript text omitted]

---

## Author Response (AR2)

5 July 2020

We would like to thank AMT and our Editor for a good peer-review process and this opportunity to submit our final copy.

Changes made to this manuscript:

- The two edits to Figure 2 caption as requested by Reviewer 2.

- Small grammatical changes for the sake of consistency as well as corrections to minor errors we detected during the final proof read.

Sincerely,
Nadia Smith

[revised manuscript text omitted]